# Koopman-informed recurrent neural networks

**Erik Lien Bolager** [1,2]**, Ana Čukarska** [1,2]**, Iryna Burak** [1,2]
**Zahra Monfared** [3,4]**, Felix Dietrich** [1,2*]
[1] *Munich Data Science Institute*
[2] *School of Computation, Information and Technology, Technical University of Munich, Germany*
[3] *Interdisciplinary Center for Scientific Computing (IWR)*
[4] *Department of Mathematics and Computer Science, Heidelberg University, Germany*

**Reviewed on OpenReview:** *https://openreview.net/forum?id=KHsnxKYG6k*

## Abstract

Recurrent neural networks are a successful neural architecture for many time-dependent problems, including time series analysis, forecasting, and modeling of dynamical systems. In the context of dynamical systems, training with backpropagation through time can lead to challenges arising from exploding or vanishing gradients. In this contribution, we introduce Koopman-informed recurrent neural networks, a computational approach to construct all weights and biases of a recurrent neural network without using gradient-based methods. The approach is based on a combination of random feature networks and Koopman operator theory for dynamical systems. The hidden parameters of a single recurrent block are sampled at random, while the outer weights are constructed using extended dynamic mode decomposition. This approach alleviates some problems with backpropagation commonly related to recurrent networks. The connection to Koopman operator theory also allows us to start using results in this area to analyze recurrent neural networks. In computational experiments on time series, forecasting for chaotic dynamical systems, control problems, and on real-world data, we observe that with comparable forecasting accuracy, the training time of the Koopman-informed recurrent neural networks is significantly improved when compared to models trained with commonly used gradient-based methods.

## 1 Introduction

Dynamical systems are defined through an evolution operator which describes how states evolve over time. In many cases one does not have access to the evolution operator but only to a certain number of observations from these systems. To identify the underlying dynamics, data-driven approaches have been proposed. One approach is to use a recurrent neural network (RNN) (Funahashi & Nakamura, 1993), which is a neural network trained to model sequential data. RNNs have been successfully applied in dynamical system modeling (Kimura & Nakano, 1998; Gajamannage et al., 2023), and, in particular, piecewise linear RNNs have been studied in the context of dynamical systems (Durstewitz, 2017; Koppe et al., 2019; Brenner et al., 2022; Hess et al., 2023). However, RNNs are also notoriously difficult to train. This is because their loss gradients backpropagated in time tend to saturate or diverge during training, which is commonly referred to as the exploding and vanishing gradient problem (EVGP) (Pascanu et al., 2013; Schmidt et al., 2021). Bifurcations may also contribute to sudden jumps in the loss observed during RNN training, potentially hindering the training process severely (Doya, 1992; Eisenmann et al., 2023). In Eisenmann et al. (2023), the authors demonstrated that specific bifurcations in ReLU-based RNNs are always associated with EVGP during training. In addition, the existence of long-term memory adversely affects the learning process of RNNs (Bengio et al., 1994; Hochreiter et al., 2001; Li et al., 2021), known as the "curse of memory". Established remedies (Hochreiter & Schmidhuber, 1997; Schmidt et al., 2021) can be used to prevent gradients from vanishing, but when modeling dynamical systems, these remedies are often insufficient. For instance, in

---

*Corresponding author, `felix.dietrich@tum.de`

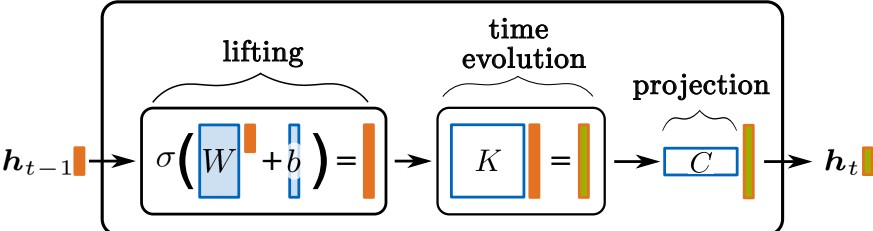

Figure 1: The central structure of an Koopman-informed RNN, where $W$ and $b$ are randomly sampled, $K$ is the approximate Koopman operator, and $\boldsymbol{h}_t$ is the state at time $t$.

systems with chaotic dynamics the gradients invariably explode, posing a challenge that cannot be mitigated just through architectural adjustments of the RNN, regularization, or constraints; instead, it is necessessary to address the problem during the training process (Mikhaeil et al., 2022).

In light of these issues, we offer a different approach, which we call *Koopman-informed RNNs.* This approach circumvents backpropagation entirely and combines the following three ideas: RNN architecture, random feature methods, and the Koopman operator. Instead of training the RNN using variants of backpropagation, we rather sample the parameters of the hidden layer(s) at random. Then, to incorporate the temporal aspect of RNNs, we use the fact that the output of the hidden layer(s) is typically in a high-dimensional space, and thus can be used to approximate the Koopman operator of the underlying dynamical system being learned. At the end, we linearly project back to the original state space. The central structure, see Figure 1, can be summarized as lifting the current state to a high-dimensional space using randomly sampled hidden layer(s), evolve one timestep forward by applying the matrix that approximates the Koopman operator, and then map the state back to the original space. If the observation and hidden spaces differ from each other, we also separately approximate a linear map from the high-dimensional hidden space to the observation space.

This choice of architecture and training process alleviates the aforementioned problems such as EVGP, and significantly speeds up the training process. It also introduces structure to the different components in an RNN, and opens up the possibility to study RNNs using tools from the Koopman literature. The usefulness of the Koopman operator stems from the fact that the evolution of the dynamical systems turns into a linear one, which implies the operator spectrum can be used to model and study the learned system (Mezić, 2005; 2013; Korda & Mezić, 2018b). In addition, the reframing of the dynamics in higher-dimensional spaces means we can apply tools from linear control theory to non-linear RNNs, such as linear-quadratic regulators (LQR).

Our idea of randomly sampling the parameters is founded on prior work by Rahimi & Recht (2008) and Barron (1993), where feedforward neural networks are treated as random feature models. In this contribution, we extend the random feature sampling algorithm by Bolager et al. (2023) to recurrent architectures. This bears a resemblance to reservoir computing (Jaeger & Haas, 2004; Lukoševičius & Jaeger, 2009), which has been successfully used to model various dynamical systems, including chaotic ones (Pathak et al., 2018; Gauthier et al., 2021). Typically, reservoir computers are constructed by drawing parameters from a fixed distribution, which is not informed by a dataset. This is one of the properties that distinguish our approach from classical reservoir computing, because our sampling procedure incorporates the training data into the definition of the parameter distribution that is being sampled from (Bolager et al., 2023). Our approach could also be seen as a novel type of reservoir architecture with data-dependent parameter distributions and additional linear structure by approximating the Koopman operator.

The ideas we present here may also be relevant for the Koopman operator community, because the approach can be rephrased in terms of a numerical algorithm for Koopman operator approximation. Many numerical approximation algorithms of the Koopman operator exist (Schmid, 2010; Williams et al., 2015; Li et al., 2017; Mezić, 2022; Schmid, 2022), and in particular, the functional space used for the approximation of the Koopman operator has been constructed with neural networks using gradient descent (Li et al., 2017) and using random features (Salam et al., 2023). Reservoir computing has also been related to Koopman operator approximation by Bollt (2021); Gulina & Mauroy (2021). To our knowledge, the relation of the Koopman

operator to the weight matrices of RNNs has not been discussed before, and is a connection we investigate further in this paper.

Our paper is organized as follows: in Section 2 we introduce the mathematical framework of Koopman-informed RNNs and discuss the application of linear control theory in our setting. In Section 3 we discuss numerical experiments to compare our approach with related models, including RNNs trained with stochastic gradient descent and reservoir computing. The experiments include approximation of stable, chaotic, and controlled systems, from both synthetic and real data. The limitations of our work and prospects for future work are discussed in Section 4.

## 2 Mathematical framework

We start by outlining the problem setting we are interested in, and define a framework of recurrent neural networks (RNNs) which we will use throughout. We then introduce parameter sampling and its use in the numerical approximation of the Koopman operator.

Let $\mathcal{X} \subseteq \mathbb{R}^{d_x}$ be an input space, $\mathcal{Y} \subseteq \mathbb{R}^{d_y}$ an output space, and $\mathcal{H} \subseteq \mathbb{R}^{d_h}$ a state space. We assume that these spaces are associated with the measures $\mu_x$, $\mu_y$, and $\mu_h$, respectively. The underlying dynamical system is then defined through the evolution operator $F$, where we may be working in an uncontrolled setting $\boldsymbol{h}_t = F(\boldsymbol{h}_{t-1})$ or a controlled setting $\boldsymbol{h}_t = F(\boldsymbol{h}_{t-1}, \boldsymbol{x}_t)$, with $\boldsymbol{h}_t \in \mathcal{H}$ and $\boldsymbol{x}_t \in \mathcal{X}$. In this paper we aim to learn dynamical systems through snapshots of the system, and denote these by $H = [\boldsymbol{h}_1, \boldsymbol{h}_2, \ldots, \boldsymbol{h}_N] \in \mathbb{R}^{d_h \times N}$ and $H' = [\boldsymbol{h}_1', \boldsymbol{h}_2', \ldots, \boldsymbol{h}_N'] \in \mathbb{R}^{d_h \times N}$. In addition, we denote the input dataset $X = [\boldsymbol{x}_1, \boldsymbol{x}_2, \ldots, \boldsymbol{x}_N] \in \mathbb{R}^{d_x \times N}$ and the dataset of observations $Y = [\boldsymbol{y}_1, \boldsymbol{y}_2, \ldots, \boldsymbol{y}_N] \in \mathbb{R}^{d_y \times N}$. In an uncontrolled system we only have access to $H$ and $H'$, and $Y$. In controlled systems we also assume access to the control states $X$. Before we go deeper into the problem setup and the datasets, and how it differs from the classical RNN setting, we will first formally define the RNN.

We denote activation functions as $\sigma \colon \mathbb{R} \to \mathbb{R}$ [1], where we are mainly working with tanh in this paper, as it is an analytic function and suitable for SWIM (Bolager et al., 2023), which is the algorithm we use to find the learnable parameters. Other functions such as ReLU are also a valid choice. The following definition outlines the models we consider.

**Definition 1.** *Let $W_h \in \mathbb{R}^{M \times d_h}$, $W_x \in \mathbb{R}^{\hat{M} \times d_x}$, $\boldsymbol{b}_h \in \mathbb{R}^M$, $\boldsymbol{b}_x \in \mathbb{R}^{\hat{M}}$, $C_h \in \mathbb{R}^{d_h \times M}$, $C_x \in \mathbb{R}^{d_h \times \hat{M}}$, $Z \in \mathbb{R}^{d_x \times d_h}$, $\boldsymbol{z} \in \mathbb{R}^{d_x}$, and $V \in \mathbb{R}^{d_y \times d_h}$. For time step $t \in \mathbb{N}_{\geq 1}$ and $\boldsymbol{h}_0 \in \mathcal{H}$, we define a recurrent neural network (RNN) by*

$$\boldsymbol{h}_t = \sigma_{hx}(C_h \, \mathcal{F}_M(\boldsymbol{h}_{t-1}) + C_x \, \mathcal{G}_{\hat{M}}(\boldsymbol{x}_t) + \boldsymbol{b}_{hx}) \tag{1}$$

$$\boldsymbol{y}_t = V \boldsymbol{h}_t \tag{2}$$

$$\boldsymbol{x}_{t+1} = Z \boldsymbol{h}_t + \boldsymbol{z}, \tag{3}$$

*where $\mathcal{F}_M(\boldsymbol{h}_{t-1}) = \sigma(W_h \, \boldsymbol{h}_{t-1} + \boldsymbol{b}_h)$ and $\mathcal{G}_{\hat{M}}(\boldsymbol{x}_t) = \sigma(W_x \, \boldsymbol{x}_t + \boldsymbol{b}_x)$ are the hidden layers. The additional step of Equation* (3) *is only added for RNNs with a one-to-many architecture.*

*Remark* 1. For coherence with the common RNN setup, we have added $\sigma_{hx}$ as an arbitrary activation function. We choose to set $\sigma_{hx}$ as the identity function as the state space is often $\mathbb{R}^{d_h}$ and to let us solve for the last linear layer using least square solvers. Other activation functions, such as the logit, are possible as well, where one would optimize using techniques from generalized linear models.

In the experiments of this paper we assume to either have snapshots from the full state of the dynamical system, or at least partial observations of the full state which can be extended with time-delay embedding. This allows us to explicitly construct snapshots of the states $\boldsymbol{h}_t$, that we then use when we construct our RNNs. This is markedly different from how the states are treated as when optimizing through backpropagation. While both try to capture the underlying dynamics, they are not treated strictly as hidden states in the classical sense in our approach. That is, the dataset setup we use in the paper aligns with the usual perspective for dynamical systems modeling, similar to the problem setting in (Hess et al., 2023). Such a

---

[1]This is extended to map $\mathbb{R}^a \to \mathbb{R}^a$ for $a \in \mathbb{N}$, by applying it element-wise.

perspective is not directly applicable to an arbitrary RNN modeling problem, e.g., translation of natural language. This difference arises because we do not employ gradient-based optimization to construct hidden representations iteratively during training, but instead explicitly create the datasets $H$ and $H'$ that capture the underlying dynamics such that we can learn a good representation of the state $\boldsymbol{h}_t$ in the RNN defined in Equation (1). However, the difference is not as stark. To make the connection to datasets provided in a more standard RNN setup, we need to consider how $H$ and $H'$ are constructed in that setting. The input used in the burn-in phase (or encoder part) of classical RNNs is the data used in $H$, while $H'$ uses part of $H$ and $Y$ to create corresponding future states that capture the underlying dynamics. We can consider two different settings that we also use in our experiments. (1) If the initial input is the full state of the system, and $Y$ has the corresponding full state one time step ahead, then we set $H' = Y$, where then $\boldsymbol{h}'_n = F(\boldsymbol{h}_n, \boldsymbol{x}_n)$ and $\boldsymbol{h}_n \in H, \boldsymbol{x}_n \in X$. (2) If the initial input data $\boldsymbol{h}_n = [\boldsymbol{h}_n^{(1)}, \ldots, \boldsymbol{h}_n^{(T')}]$ is a partial observation with time delay $T'$, and $\boldsymbol{y}_n \in Y$ is the next partial observed state, then the corresponding $\boldsymbol{h}'_n \in H'$ equals $[\boldsymbol{y}_n, \boldsymbol{h}_n^{(1)}, \boldsymbol{h}_n^{(2)}, \ldots, \boldsymbol{h}_n^{(T'-1)},]$. After training we simply set $\boldsymbol{h}_0$ equal to the data passed as input during the burn-in phase, and then compute the output and the next state in the RNN using the previous state and input $\boldsymbol{x}_t$ (if applicable), exactly the same way as in a regular RNN setting. In the computational experiments we consider in this paper the construction of the hidden states is quite straightforward, but there may be applications where this construction must be more elaborate. A major benefit of the more explicit construction of the hidden states is that they are immediately more interpretable than the ones created through iterative training.

Regarding the training procedure of RNNs: the classical way to train this type of RNN is through iterative backpropagation known as backpropagation-through-time, which suffers the aforementioned issues such as EVGP and high computational complexity. To circumvent backpropagation, we instead start by sampling the hidden layer parameters, as explained next.

## 2.1 Sampling RNN

*Sampled neural networks* are neural networks where the parameters of the hidden layers are sampled from some distribution, and the last linear layer is either sampled or, more typically, constructed by solving a linear problem. Following Bolager et al. (2023), we sample the weights and biases of the hidden layers of the two networks $F_{\mathcal{H}} = C_h \mathcal{F}_M$ and $F_{\mathcal{X}} = C_x \mathcal{G}_{\hat{M}}$, by sampling pairs of points from the domain $\mathcal{H}$ and $\mathcal{X}$, and then construct the weights and biases from these pairs of points. For the final layer, we use (generalized) linear regression techniques to find the parameters based on the observations. Concretely, let $\mathbb{P}_{\mathcal{H}}$ and $\mathbb{P}_{\mathcal{X}}$ be probability distributions over $\mathcal{H}^2$ and $\mathcal{X}^2$ respectively. For each neuron in the hidden layer of $F_{\mathcal{H}}$, sample $(\boldsymbol{h}^{(1)}, \boldsymbol{h}^{(2)}) \sim \mathbb{P}_{\mathcal{H}}$ and set the weight $w$ and the bias $b$ of said neuron to

$$\boldsymbol{w} = s_1 \frac{\boldsymbol{h}^{(2)} - \boldsymbol{h}^{(1)}}{\|\boldsymbol{h}^{(2)} - \boldsymbol{h}^{(1)}\|^2}, \quad b = -\langle \boldsymbol{w}, \boldsymbol{h}^{(1)} \rangle + s_2, \tag{4}$$

where $\|\cdot\|$ and $\langle \cdot, \cdot \rangle$ are the Euclidean norm and inner product, and $s_1, s_2 \in \mathbb{R}$ are constants depending on the choice of activation function. In this paper, we only train networks with one hidden layer. Regarding multilayer sampling we direct the reader to Bolager et al. (2023), where the authors include the full sample and construction procedure for an arbitrary number of hidden layers.

By using the parameter sampling technique above, we can adapt the weights and biases to the underlying domain and construct weights with direction along the data. Empirically, we will demonstrate that this leads to improvements over using data-agnostic distributions such as the standard Gaussian. One can choose arbitrary probability distributions as $\mathbb{P}_{\mathcal{H}}$ and $\mathbb{P}_{\mathcal{X}}$, with uniform distribution being a common choice. For the supervised setting, Bolager et al. (2023) proposed a sampling distribution that is constructed to have a high density at the steepest gradients of the target function. For this paper, we sample with densities $p_{\mathcal{H}}$ and $p_{\mathcal{X}}$ proportional to

$$p_{\mathcal{H}} \propto \frac{\|F(\boldsymbol{h}^{(2)}) - F(\boldsymbol{h}^{(1)})\|}{\|\boldsymbol{h}^{(2)} - \boldsymbol{h}^{(1)}\|}, \quad p_{\mathcal{X}} \propto 1, \tag{5}$$

respectively. In practice, we do not have access to the full state space $\mathcal{H}$, and we therefore discretize using datasets $H$ and $H'$, with Equation (5) defining our probability mass functions.

Once the weights and biases are constructed, we only need to solve the following optimization problem,

$$[C_h, C_x] = \underset{\hat{C}_h, \hat{C}_x}{\arg\min} \sum_{n=1}^{N} \|(\hat{C}_h \mathcal{F}_M(\boldsymbol{h}_n) + \hat{C}_x \mathcal{G}_{\hat{M}}(x_n) - \boldsymbol{h}'_n\|^2 \tag{6}$$

$$= \underset{\hat{C}_h, \hat{C}_x}{\arg\min} \sum_{n=1}^{N} \|(\hat{C}_h \, \sigma(W_h \, \boldsymbol{h}_n + \boldsymbol{b}_h) + \hat{C}_x \, \sigma(W_x \, \boldsymbol{x}_n + \boldsymbol{b}_x) + \boldsymbol{b}_{hx}) - \boldsymbol{h}'_n\|^2.$$

To summarize, we define a *sampled RNN* as a model that is constructed by sampling weights of the hidden layer of the RNN and subsequently solving the regression problem in Equation (6).

## 2.2 Involving the Koopman operator

We now introduce the Koopman operator and incorporate it into our sampled RNNs. We do this to both add more structure and interpretability to the matrices $C_h$ and $C_x$, allow for linear control theory later on, and apply analysis from the Koopman literature to better understand RNNs.

Let us first give a brief introduction of the Koopman operator. Given a suitable function(al) space $\mathcal{F}$, the Koopman operator $\mathcal{K}: \mathcal{F} \to \mathcal{F}$ is defined as

$$[\mathcal{K}\phi](\boldsymbol{h}) = (\phi \circ F)(\boldsymbol{h}), \quad \phi \in \mathcal{F}, \boldsymbol{h} \in \mathcal{H}.$$

This operator captures the evolution of the dynamical system in the function space $\mathcal{F}$ instead of the state space. In most cases, this adds complexity because $\mathcal{F}$ is infinite-dimensional, but has the benefit that the action of the operator is always linear—even for strongly nonlinear evolutions $F$. This allows the study of nonlinear dynamical systems by looking at the spectrum of the associated linear operator. More specifically, assume there exist eigenfunctions $\varphi_k$ of $\mathcal{K}$ with corresponding eigenvalues $\lambda_k$. Then, for an arbitrary function $\phi_i \in \text{Span}\{\varphi_k\}$ we have that

$$\phi(\boldsymbol{h}_{t+1}) = \mathcal{K} \, \phi(\boldsymbol{h}_t) = \mathcal{K} \sum_k c_k \, \varphi_k(\boldsymbol{h}_t) = \sum_k c_k \, \mathcal{K} \, \varphi_k(\boldsymbol{h}_t) = \sum_k c_k \, \lambda_k \, \varphi_k(\boldsymbol{h}_t),$$

where $c_k$ are the so-called *Koopman modes* associated with $\phi$. Both the Koopman modes and the eigenvalues can be useful for analyzing the dynamical system, which we demonstrate further in Section 3.5. For a more extensive introduction to the Koopman operator and surrounding theory, see Appendix A.

The estimation procedure we use to approximate the Koopman operator is called *extended dynamic mode decomposition (EDMD)*, which is an algorithm to construct a finite-dimensional approximation of the operator. We give a very brief description of EDMD here and give a more thorough introduction in Appendix A.1 and Appendix A.2. In the uncontrolled setting, the EDMD method requires a predetermined dictionary $\Psi_M = \{\psi_1, \ldots, \psi_M | \psi_i : \mathcal{H} \to \mathbb{R}\} \subset \mathcal{F}$. In this paper, we choose the randomly sampled neurons as the dictionary functions and call this particular dictionary $\mathcal{F}_M$. Then we approximate the Koopman operator $\mathcal{K}$ in this subspace $\mathcal{F}_M$ using the data $H, H'$, by minimizing

$$K = \underset{\tilde{K} \in \mathbb{R}^{M \times M}}{\arg\min} \sum_{n=1}^{N} \|\mathcal{F}_M(\boldsymbol{h}'_n) - \tilde{K}\mathcal{F}_M(\boldsymbol{h}_n)\|, \quad \boldsymbol{h}'_n \in H', \boldsymbol{h}_n \in H.$$

Defining $\mathcal{F}_M(H) = [\mathcal{F}_M(\boldsymbol{h}_1), \ldots, \mathcal{F}_M(\boldsymbol{h}_N)] \in \mathbb{R}^{M \times N}$ and $\mathcal{F}_M(H') = [\mathcal{F}_M(\boldsymbol{h}'_1), \ldots, \mathcal{F}_M(\boldsymbol{h}'_N)] \in \mathbb{R}^{M \times N}$, the solution to the minimization problem can then be written as

$$K = \mathcal{F}_M(H') \, \mathcal{F}_M(H)^+, \tag{7}$$

where $^+$ is the matrix pseudoinverse. Similarly, in the controlled setting, the approximation involves two separate matrices $K$ and $B$,

$$[K, B] = \mathcal{F}_M(H') \, (\mathcal{F}_M(H) \oplus \mathcal{G}_{\hat{M}}(X))^+,$$

where $\mathcal{G}_{\hat{M}}$ is the second dictionary mapping from $\mathbb{R}^{d_x}$ to $\mathbb{R}^{\hat{M}}$, and $(\mathcal{F}_M(H) \oplus \mathcal{G}_{\hat{M}}(X)) \in \mathbb{R}^{(M+\tilde{M}) \times N}$ is the concatenation of the matrices. Regardless of whether uncontrolled or controlled, we compute a mapping $C$, which projects from the high dimensional dictionary space back to the state space, by minimizing $\|H - C\mathcal{F}_M(H)\|$, with the solution

$$C = H\mathcal{F}_M(H)^+.$$

To compute trajectories using the approximations $K \in \mathbb{R}^{M \times M}$, $B \in \mathbb{R}^{M \times \hat{M}}$, and $C \in \mathbb{R}^{d_h \times M}$, we have

$$\boldsymbol{h}_t = C(K\sigma(W_h \boldsymbol{h}_{t-1} + \boldsymbol{b}_h) + B\sigma(W_x \boldsymbol{x}_t + \boldsymbol{b}_x)). \tag{8}$$

It is important to notice that the resulting function in Equation (8) consists of two neural networks applied to the previous state and input. The hidden layers are sampled, and the outer matrices are constructed using linear solvers. After computing the different matrices $C$, $K$, and $B$, we can always collapse the matrix products into $C_h = CK$ and $C_x = CB$. Hence, Equation (8) follows Definition 1, and in particular, is a *sampled recurrent neural network* per our definition. In the rest of the paper, we will turn our focus solely on the architecture specified in Equation (8), which we refer to as *Koopman-informed RNN* (KIRNN).

To motivate further the choice of including the Koopman operator by splitting the matrices $C_h$ and $C_x$ as described, we give the following reason: the hidden layers $\mathcal{F}_M$ and $\mathcal{G}_{\hat{M}}$ map their respective input to a higher dimensional space. In a higher dimensional space, the possibly nonlinear evolution described by $F$ becomes more and more linear (Korda & Mezić, 2018b). This evolution is then captured by $K$ and $B$ before we map down to the state space through $C$. The matrix $K$ is then an approximation of the Koopman operator of the system we want to learn. Choosing $M$ large enough means we can capture the evolution through a linear map before we map back to the state space. By extending the Koopman theory to the controlled setting, one also finds a similar interpretation of $B$ (Korda & Mezić, 2018a).

## 2.3 Nonlinear optimal control with RNNs

The goal of optimal control is to find a sequence of controls $\{\boldsymbol{x}_t\}$ that minimize a cost function $J(\{\boldsymbol{h}_t\}, \{\boldsymbol{x}_t\})$ and steer the system towards a chosen state $\boldsymbol{h}^*$, where $\boldsymbol{h}_t = F(\boldsymbol{h}_{t-1}, \boldsymbol{x}_t)$. For the purpose of this paper, we assume $J$ is quadratic. When the underlying dynamics described by $F$ is nonlinear, tools such as nonlinear model predictive control (MPC) are necessary (Mayne et al., 2000; Grüne & Pannek, 2017). However, because we established the connection of our network architecture and the Koopman operator, we can use tools from linear control theory to control the nonlinear system $F$ using KIRNN. For our purposes we opt to use linear-quadratic regulator (LQR) when computing $\boldsymbol{x}_t$, and we now show how to incorporate this controller into KIRNN. Given the cost matrices $Q, R, S$, the quadratic cost function we work with is

$$J(\boldsymbol{h}_t, \boldsymbol{x}_t) = \sum_{t=0}^{\infty} (\boldsymbol{h}_t^\mathsf{T} Q\boldsymbol{h}_t + \boldsymbol{x}_{t+1}^\mathsf{T} R\boldsymbol{x}_{t+1} + 2\boldsymbol{h}_t^\mathsf{T} S\boldsymbol{x}_{t+1}).$$

The optimal control sequence $\{\boldsymbol{x}_t\}_{t=1}^{\infty}$ can be found by first setting

$$\hat{Z} = (R + B^\mathsf{T} PB)^+ (B^\mathsf{T} PK + S^\mathsf{T}),$$

where $P$ is the solution to the discrete-time algebraic Riccati equation

$$P = K^\mathsf{T} PK - \left[ (KPB + S)(R + B^\mathsf{T} PB)^+ (B^\mathsf{T} PK + S^\mathsf{T}) \right] + Q.$$

The matrices $K$ and $B$ are determined by using EDMD with our sampled hidden layers $\mathcal{F}_M$ and $\mathcal{G}_{\hat{M}}$ as the dictionary. We then find the optimal control $\hat{\boldsymbol{x}}_{t+1} = -\hat{Z}(\boldsymbol{h}_t - \boldsymbol{h}^*)$. Notice that $\hat{\boldsymbol{x}}_{t+1} \in \mathbb{R}^{\hat{M}}$, and not in the control input space $\mathcal{X}$. Before we can evolve to the next state $\boldsymbol{h}_t$ we need to project down to the controlled input space $\mathcal{X}$. Let $\hat{C} = X[\mathcal{G}_{\hat{M}}(X)]^+$. Then we have

$$\begin{aligned}
\boldsymbol{h}_t &= C(K\mathcal{F}_M(\boldsymbol{h}_{t-1}) + B\mathcal{G}_{\hat{M}}(\boldsymbol{x}_t)), \\
\boldsymbol{x}_{t+1} &= Z\boldsymbol{h}_t - z = -\hat{C}\,\hat{Z}(\boldsymbol{h}_t - \boldsymbol{h}^*), \\
\boldsymbol{y}_t &= V\boldsymbol{h}_t.
\end{aligned} \tag{9}$$

Equation (9) thus shows how one can solve a nonlinear optimal control problem using linear control of KIRNNs.

All the methods above can then be summarized by sampling weights in Algorithm 1, constructing the RNN in Algorithm 2, and nonlinear control in Algorithm 3. The computational complexity of the full algorithm is dominated by the number of neurons $M$ for the network $\mathcal{F}_M$ acting on the state and $\hat{M}$ for the network $\mathcal{G}_{\hat{M}}$ acting on the control, assuming $M, \hat{M} \geq \max\{d_h, d_x\}$. As the sampling part of the proposed algorithm is efficient, the bottleneck is rather the inverse operations when computing $C$, $K$, $B$, $Z$, and $V$. They are cubic in terms of $M$ and $\hat{M}$, where the size of $M$ dominates cubically for $C$, $K$, and $V$, while $\hat{M}$ dominates cubically the computation for $B$ and $Z$. As the training procedure for the parameters of the sampled RNN is not iterative, these fairly expensive least-squares computations only need to be done once. In addition, if a problem requires to sample many neurons, meaning $M$ or $\hat{M}$ is large, algorithms concerning pseudoinverse and least square solutions have been studied extensively (Meng et al., 2014).

---

**Algorithm 1** Sampling weights and bias for a given dataset and probability distribution.

---

**procedure** SAMPLE-LAYER($Z, \mathbb{P}_{\mathcal{Z}}$)
    $W_z \in \mathbb{R}^{M \times d_z}, \boldsymbol{b}_z \in \mathbb{R}^{d_z}$
    **for** $j = 1, 2, \ldots M$ **do**
        Sample $(\boldsymbol{z}^{(1)}, \boldsymbol{z}^{(2)}) \sim \mathbb{P}_z$ from $Z \times Z$
        $W_z^{[j,:]} = \frac{\boldsymbol{z}^{(2)} - \boldsymbol{z}^{(1)}}{\|\boldsymbol{z}^{(2)} - \boldsymbol{z}^{(1)}\|^2}^{\mathsf{T}}$
        $\boldsymbol{b}_z^{[j]} = -\langle (W_z^{[j,:]})^{\mathsf{T}}, \boldsymbol{z}^{(1)} \rangle$
    **end for**
    Return $W_z, \boldsymbol{b}_z$
**end procedure**

---

**Algorithm 2** Constructing KIRNNs for controlled setting.

---

**procedure** SAMPLE-RNN($X, Y, H, H'$)
    $W_x, \boldsymbol{b}_x \leftarrow$ SAMPLE-Layer($X, \mathbb{P}_X$)
    $W_h, \boldsymbol{b}_h \leftarrow$ SAMPLE-Layer($H, \mathbb{P}_H$)
    $\mathcal{F}_M(\cdot), \mathcal{G}_{\hat{M}}(\cdot) \leftarrow \sigma(W_h \cdot + \boldsymbol{b}_h), \sigma(W_x \cdot + \boldsymbol{b}_x)$
    $[K, B] = \mathcal{F}_M(H')(\mathcal{F}_M(H) \oplus \mathcal{G}_{\hat{M}}(X))^+$
    $C = H \mathcal{F}_M(H)^+$
    $V = Y H^+$
    Return $V, C, K, \mathcal{F}_M, B, \mathcal{G}_{\hat{M}}$
**end procedure**

---

**Algorithm 3** Nonlinear optimal control of system $F$ using LQR and KIRNN, with cost matrices $Q, R, S$. The procedure combines fitting the model, and prediction of the trajectory for initial condition $\boldsymbol{h}_0$, target state $\boldsymbol{h}^*$, and $t = 1, 2, \ldots, T$.

---

**procedure** NONLINEAR-CONTROL($X, Y, H, H', \mathrm{T}, \boldsymbol{h}_0, \boldsymbol{h}^*$)
    $V, C, K, \mathcal{F}_M, B, \mathcal{G}_{\hat{M}} \leftarrow$ SAMPLE-RNN($X, Y, H, H'$)
    $\tilde{P} \leftarrow (R + B^{\mathsf{T}} P B)^+$
    $P \leftarrow K^{\mathsf{T}} P K - \left[ (KPB + S)\tilde{P}(B^{\mathsf{T}} PK + S^{\mathsf{T}}) \right] + Q$
    $Z \leftarrow \tilde{P}(B^{\mathsf{T}} PK + S^{\mathsf{T}})$
    $\hat{C} \leftarrow X[\mathcal{G}_{\hat{M}}(X)]^+$
    $\boldsymbol{x}_1 \leftarrow -\hat{C} Z(\boldsymbol{h}_0 - \boldsymbol{h}^*)$
    **for** $t = 1, 2, \ldots, \mathrm{T}$ **do**
        $\boldsymbol{h}_t \leftarrow C(K\mathcal{F}_M(\boldsymbol{h}_{t-1}) + B\mathcal{G}_{\hat{M}}(\boldsymbol{x}_t))$
        $\boldsymbol{x}_{t+1} \leftarrow -\hat{C} \hat{Z}(\boldsymbol{h}_t - \boldsymbol{h}^*)$
        $\boldsymbol{y}_t = V\boldsymbol{h}_t$
    **end for**
    Return $\{\boldsymbol{x}_t, \boldsymbol{h}_t, \boldsymbol{y}_t\}_{t=1}^T$
**end procedure**

---

## 2.4 The role of nonlinearity

The role of the activation function is to provide nonlinearity to neural networks, which is crucial to obtain universal approximation. In KIRNNs, the approximation of the Koopman operator also requires a nonlinear dictionary to be accurate. Currently, from our Definition 1, we see that each state is lifted to a higher-dimensional space followed by an activation function, for every timestep $t \in \mathbb{N}_{>0}$. By extending results from Koopman theory, we can shed light on the approximation error of the network over several time steps.

We start by defining $L^2 := L^2(\mathcal{H}, \mu_h)$ and stating the required assumptions for our results.

**Assumption 1.** The assumptions on $\mu_h$, $\mathcal{F}_M$, $\mathcal{K}$, and the underlying system $F$ are the following.

    1.1. $\mu_h$ is regular and finite for compact subsets.

    1.2. The hidden layer $\mathcal{F}_M$ satisfies $\mu_h\{\boldsymbol{h} \in \mathcal{H} \mid \boldsymbol{c}^\mathsf{T}\mathcal{F}_M(\boldsymbol{h}) = 0\} = 0$, for all nonzero $\boldsymbol{c} \in \mathbb{R}^M$.

    1.3. The Koopman operator $\mathcal{K}: L^2 \to L^2$ is a bounded operator.

The first two points are not very restrictive and hold for many measures and activation functions, including the *tanh* activation function and the Lebesgue measure. It is worth noting that ReLU does not satisfy Assumption 1, but we suspect continuous versions of ReLU can be shown to satisfy it; see Lemma 1. The third assumption is common in Koopman approximation theory for showing convergence. It holds for a broad set of dynamical systems (See Appendix B.1 for further discussion of all three points). Finally, we also require $\mathcal{H}$ to follow Definition 2 in Appendix B.1, which allows the use of universal approximation theory for the weight space restricted to $\mathcal{X} \times \mathcal{X}$ Bolager et al. (2023).

We now denote $L^2_{d_y}$ as the space of vector valued functions functions $f = [f_1, f_2, \ldots, f_{d_y}]$, where $f_i \in L^2$ and $\|f\|_{L^2_{d_y}} = \sum_{i=1}^{d_y}\|f_i\|_{L^2}$. We let $F^t(\boldsymbol{h}_0) = \boldsymbol{h}_t$ be the true state after time $t$, and $K_N$ be the solution of Equation (7), where $N$ data points have been used to solve the least square problem.

**Theorem 2.** *Let $f \in L^2_{d_y}$, $H, H'$ be the dataset with $N$ data points used in Equation (7), and Assumption 1 holds. In addition we assume $\mathcal{H}$ follows Definition 2. Then for any $\epsilon > 0$ and $T \in \mathbb{N}$, there exist an $M \in \mathbb{N}$ and hidden layers $\mathcal{F}_M$ and matrices $C$ such that*

$$\lim_{N\to\infty} \int_{\mathcal{H}} \|CK_N^t\mathcal{F}_M - f \circ F^t\|_2^2 d\mu_h < \epsilon,$$

*for all $t \in [1, 2, \ldots, T]$.*

*Proof.* We give an outline of the proof here, while the full proof can be found in Appendix B.1.

- In Lemma 1, we show how parts of the assumption made are satisfied by a large class of activation functions, which include *tanh*.

- In Lemma 3, we loosen some of the assumptions made in previous EDMD convergence results (Korda & Mezić, 2018a), so that they apply to neural networks.

- Finally, in Theorem 5 we combine the results of Korda & Mezić (2018a) for EDMD convergence and universal approximation to produce the result above. Here, the universality property of neural networks is used to make sure $CF_M$ can approximate $f$, while the approximation of the underlying dynamics is done through convergence of $K_N$ to $\mathcal{K}$.

$\square$

For prediction of the system output, as the identity function $Id(\boldsymbol{h}) \mapsto \boldsymbol{h}$ is in $L^2_{d_h}$, the result above implies convergence of $\int_{\mathcal{H}}\|CK_N^t\mathcal{F}_M - F^t\|_2^2 d\mu_h$. In Appendix B.2 we discuss the limitations of the result w.r.t. the controlled setting.

The result above shows that nonlinearity is only required to create a high-dimensional representation of the initial state; once that is achieved, the resulting RNN can be a fully linear model. This may open up avenues for combining results from linear RNNs (Li et al., 2021), to show similar results for larger function classes. The results, interestingly, do not include ReLU activation functions, due to the assumptions made in Assumption 1, and it is unclear whether it is easy to extend the results to include ReLU. It is also worth noting that the results provide an insight into the learning algorithm itself, and future work may look into

how to extend Theorem 2 to a probabilistic result that takes into account also the sampling of the network parameters, similar to results for feed-forward networks (Rudi & Rosasco, 2017).

Empirical results reveal more nuance to the theoretical result. For many systems with a limited number of neurons and data points, projecting to the state space and then lifting it through the hidden layers *in each iteration of the KIRNN* improves accuracy and stability, compared to simply applying $CK^t$ to the initial condition to reach the state at time $t$. This has been observed both in our experiments and in the EDMD literature (Constante-Amores et al., 2024). It remains unclear exactly why this is the case, but we can make a few observations. Consider the $d_h$-dimensional manifold induced by the hidden layer, namely $\mathcal{M} := \mathcal{F}_M(\mathcal{H}) \subset \mathbb{R}^M$. Firstly, when the Koopman operator is applied elementwise to the hidden layer $\mathcal{F}_M$ and then evaluated at $\boldsymbol{h} \in \mathcal{H}$, by definition we have $[\mathcal{K}\mathcal{F}_M](\boldsymbol{h}) \in \mathcal{M}$. However, the mapping from $\mathcal{F}_M(\boldsymbol{h})$ to $[\mathcal{K}\mathcal{F}_M](\boldsymbol{h})$ may not be linear in the $M$-dimensional subspace, because the linearity of $\mathcal{K}$ only holds in the infinite dimensional function space. The truncation to $M$ functions and using the approximation $K$ instead may then move $\mathcal{F}_M(\boldsymbol{h})$ away from $\mathcal{M}$. By projecting down to the state space and mapping it back to $\mathcal{M}$, we at least guarantee that the point is on the manifold, even if the map $\mathcal{F}_M \circ C$ is not the optimal one. Intuitively, the bigger the difference between the manifold dimension $d_h$ and $M$ is, the smaller the difference between $CK^2\mathcal{F}_M(\boldsymbol{h})$ and $CK\mathcal{F}_M(CK\mathcal{F}_M(\boldsymbol{h}))$ due to the fact that there are more orthogonal directions w.r.t. $\mathcal{M}$ in $\mathbb{R}^M$. However, the exact relationship, as well as how the geometry of $\mathcal{M}$ affects the necessity of mapping down to $\mathcal{M}$ at each timestep, remains unclear. Considering all the facts above, we find that following Definition 1, and in particular, Equation (9), is beneficial for the number of neurons and size of dataset we are working with.

## 3 Computational experiments

We now discuss a series of experiments designed to illustrate the benefits and challenges of our approach and compare with existing approaches. The state-of-the-art recurrent architecture for modeling dynamical systems is shPLRNN by Hess et al. (2023). Due to the similarities our method bears with reservoir models, we also compare with an established reservoir model, namely an echo state network (ESN). Both of these are explained in Section 3.1.

In every experiment in this section we assume the datasets $H$ and $H'$ consist of the full state $\boldsymbol{h}$, except for the experiment in Section 3.3, where we only assume that partial observations of the state are available, and Section 3.7, where it is unknown if the full state is observed for these real-world examples. Hyperparameters for all models are given in Appendix D, the evaluation metrics are explained in Appendix C and a further discussion as well as the hardware details are provided in Appendix D. The code to reproduce the experiments from the paper, as well as an up-to-date code base, is accessible at:



https://gitlab.com/fd-research/kirnn-paper
https://gitlab.com/fd-research/kirnn.



In Table 1, we list the main quantitative results for most of the experiments. Each reported result stems from five different runs, where the random seed is changed in order to ensure a more robust result. We give the mean over these five runs, as well as the minimum and maximum among them.

### 3.1 Related Architectures

### 3.1.1 SOTA for Dynamical Systems Modeling: ReLU-based RNNs

Different RNN architectures to model dynamical systems, in particular chaotic ones, have been proposed, and with both theoretical and empirical backing. We start by a brief review of some key ReLU-based models.

A piecewise linear RNN (PLRNN), introduced by Koppe et al. (2019), has the generic form

$$\boldsymbol{h}_t = W_h^{(1)} \boldsymbol{h}_{t-1} + W_h^{(2)} \sigma(\boldsymbol{h}_{t-1}) + \boldsymbol{b}_0 + W_x \boldsymbol{x}_t, \tag{10}$$

where $\sigma(\boldsymbol{h}_{t-1}) = \max(0, \boldsymbol{h}_{t-1})$ is the element-wise rectified linear unit (ReLU) function, $W_h^{(1)} \in \mathbb{R}^{d_h \times d_h}$ is a diagonal matrix of auto-regression weights, $W_h^{(2)} \in \mathbb{R}^{d_h \times d_h}$ is a matrix of connection weights, the vector

Table 1: Results from computational experiments. We report the training time and MSE (mean squared error) or EKL (empirical Kullback–Leibler divergence, see Appendix C) for a KIRNN (our approach), a reservoir model ESN (see Section 3.1.2) and a state of the art backpropagation-based RNN called shPLRNN (see Section 3.1.1).

| Example | Model | Time [s]: avg (min, max) | MSE: avg (min, max) |
|---|---|---|---|
| Van der Pol | KIRNN | 0.26 (0.18, 0.35) | 9.55e-4 (7.08e-4, 1.28e-3) |
| | ESN | 3.76 (2.29, 5.40) | 1.58e-2 (1.15e-2, 2.07e-2) |
| | shPLRNN | 217.93 (203.10, 251.51) | 1.39e-2 (5.66e-3, 3.00e-2) |
| 1D Van der Pol | KIRNN | 0.29 (0.25, 0.31) | 5.06e-3 (1.57e-4, 1.58-2) |
| Weather (day) | KIRNN | 4.87 (4.83, 4.92) | $2.239°C$ ($2.088°C$, $2.392°C$) |
| | ESN | 2.17 (2.12, 2.20) | $3.323°C$ ($2.867°C$, $3.962°C$) |
| | shPLRNN | 242.41 (225.74, 269.35) | $2.174°C$ ($1.803°C$, $2.548°C$) |
| Weather (week) | KIRNN | 4.87 (4.81, 4.90) | $4.624°C$ ($4.169°C$, $4.867°C$) |
| | ESN | 3.49 (3.48, 3.50) | $7.238°C$ ($6.144°C$, $8.110°C$) |
| | shPLRNN | 268.6 (259.46, 275.80) | $5.412°C$ ($5.104°C$, $5.861°C$) |
| Electricity consumption | KIRNN | 3.17 (3.11, 3.28) | $1.66V$ ($1.61V$, $1.70V$) |
| | ESN | 3.31 (3.27, 3.33) | $2.418V$ ($2.239V$, $2.595V$) |
| | shPLRNN | 1330.25 (1226.26, 1443.06) | $7.944V$ ($7.927V$, $7.968V$) |
| | | | EKL: avg (min, max) |
| Lorenz-63 | KIRNN | 1.67 (1.34, 1.92) | 4.36e-3 (3.66e-3, 5.36e-3) |
| | ESN | 3.54 (2.87, 4.47) | 8.73e-3 (7.20e-3, 1.06e-2) |
| | shPLRNN | 607.42 (581.39,650.56) | 5.79e-3(4.41e-3,7.56e-3) |
| Rössler | KIRNN | 5.36 (4.39, 6.39) | 1.57e-4 (5.86e-5, 3.82e-4) |
| | ESN | 8.11 (7.94, 8.31) | 8.33e-5 (3.79e-5, 2.25e-4) |
| | shPLRNN | 866.17 (848.56, 939.06) | 6.53e-4 (4.35e-4,1.09e-3) |

$\boldsymbol{b}_0 \in \mathbb{R}^{d_h}$ represents the bias, and the external input is weighted by $W_x \in \mathbb{R}^{d_h \times d_x}$. Brenner et al. (2022) extended this basic structure by incorporating a linear spline basis expansion, referred to as the dendritic PLRNN (dendPLRNN)

$$\boldsymbol{h}_t = W_h^{(1)} \boldsymbol{h}_{t-1} + W_h^{(2)} \sum_{j=1}^{J} \alpha_j \, \sigma(\boldsymbol{h}_{t-1} - \boldsymbol{b}_j) + \boldsymbol{b}_0 + W_x \boldsymbol{x}_t, \tag{11}$$

where $\{\alpha_j, \boldsymbol{b}_j\}_{j=1}^{J}$ represents slope-threshold pairs, with $J$ denoting the number of bases. This expansion was introduced to increase the expressivity of each unit's nonlinearity, thereby facilitating dynamical systems modeling in reduced dimensions. Moreover, Hess et al. (2023) proposed the following "1-hidden-layer" ReLU-based RNN, which they referred to as the shallow PLRNN (shPLRNN)

$$\boldsymbol{h}_t = W_h^{(1)} \boldsymbol{h}_{t-1} + W_h^{(2)} \sigma(W_h^{(3)} \boldsymbol{h}_{t-1} + \boldsymbol{b}_1) + \boldsymbol{b}_0 + W_x \boldsymbol{x}_t, \tag{12}$$

where $W_h^{(1)} \in \mathbb{R}^{d_h \times d_h}$ is a diagonal matrix, $W_h^{(2)} \in \mathbb{R}^{d_h \times M}$ and $W_h^{(3)} \in \mathbb{R}^{M \times d_h}$ are rectangular connectivity matrices, and $\boldsymbol{b}_1 \in \mathbb{R}^{M}$, $\boldsymbol{b}_0 \in \mathbb{R}^{d_h}$ denote thresholds. The combination of Generalized Teacher Forcing (GTF) and shPLRNN results in a powerful dynamical system modeling algorithm on challenging real-world data; for more information see Hess et al. (2023). We also note that when $M > d_h$, it is possible to rewrite any shPLRNN as a dendPLRNN by expanding the activation of each unit into a weighted sum of ReLU nonlinearities (Hess et al., 2023).

Finally, a clipping mechanism can be added to the shPLRNN to prevent states from diverging to infinity as a result of the unbounded ReLU nonlinearity

$$\boldsymbol{h}_t = W_h^{(1)}\boldsymbol{h}_{t-1} + W_h^{(2)}\big[\sigma(W_h^{(3)}\boldsymbol{h}_{t-1} + \boldsymbol{b}_1) - \sigma(W_h^{(3)}\boldsymbol{h}_{t-1})\big] + \boldsymbol{b}_0 + W_x\boldsymbol{x}_t. \tag{13}$$

This guarantees bounded orbits under certain conditions on the matrix $W_h^{(1)}$ (Hess et al., 2023).

In our experiments, we use the clipped shPLRNN trained by GTF and compare it to our approach. KIRNN differs from the above mentioned methods in multiple ways: the training mechanism we use is not iterative, thus it typically requires less computation time; furthermore, we reach a linearized representation due to the Koopman connection, which allows the use of linear control methods.

### 3.1.2 Reservoir Models: Echo State Networks

As our newly proposed method bears similarities to a reservoir computing architecture, we have also trained reservoir models as part of our computational experiments. We used the echo state network (ESN) introduced by Jaeger & Haas (2004); here we briefly introduce the main ideas behind ESNs, and we refer the interested reader to the review by Lukoševičius & Jaeger (2009) for more details.

An ESN consists of a reservoir and a readout. The reservoir contains neurons which are randomly connected to inputs and these are not trained, only initialized. Denoting the inputs as $\boldsymbol{h}_t \in \mathbb{R}^{N_h}$ and output as $\boldsymbol{y}_t \in \mathbb{R}^{N_y}$, and the internal reservoir states as $\boldsymbol{k}_t \in \mathbb{R}^{N_k}$, the reservoir provides an update rule for the internal units as

$$\boldsymbol{k}_{t+1} = \sigma\left(W^{in}\boldsymbol{h}_{t+1} + W\boldsymbol{k}_t + W^{back}\boldsymbol{y}_t\right), \tag{14}$$

for an activation function $\sigma$ and weight matrices $W^{in} \in \mathbb{R}^{N_k \times N_h}$, $W \in \mathbb{R}^{N_k \times N_k}$ and $W^{back} \in \mathbb{R}^{N_k \times N_y}$. After the reservoir comes the readout, which maps the inputs, reservoir states, and outputs to a new output state

$$\boldsymbol{y}_{t+1} = \sigma_{out}\left(W^{out}(\boldsymbol{h}_{t+1} \oplus \boldsymbol{k}_{t+1} \oplus \boldsymbol{y}_t)\right),$$

where $\sigma^{out}$ is the output activation, $W^{out} \in \mathbb{R}^{N_y \times N_y}$ are output weights and $\boldsymbol{h}_{t+1} \oplus \boldsymbol{k}_{t+1} \oplus \boldsymbol{y}_t$ denotes the concatenation of $\boldsymbol{h}_{t+1}$, $\boldsymbol{k}_{t+1}$, and $\boldsymbol{y}_t$. In the readout the model learns the connections from the reservoir to the readout, for example via (regularized) regression. A so-called feedback connection allows for the readout values to be fed back into the reservoir, as shown in Equation (14), establishing a recurrent relation.

In the context of reservoir models, the stability and expressivity of the model are often discussed. The stability is related to the so-called *echo state property* (Jaeger & Haas, 2004) which is a stability condition, which is typically satisfied if the reservoir weights are contractive, for example in the case of *tanh* activation (Lukoševičius & Jaeger, 2009). The expressivity is often related to processes evolving over different time scales; one way to control this is by a *leak rate* parameter $\alpha$ which determines what proportion of the information from the previous state is passed on to the next state

$$\boldsymbol{k}_{t+1} = (1-\alpha)\boldsymbol{k}_t + \alpha f\left(W^{in}\boldsymbol{h}_{t+1} + W\boldsymbol{k}_t + W^{back}\boldsymbol{y}_t\right).$$

The main strength of the reservoir computer is its very fast training time, which opens the door to neural architecture search (Strubell et al., 2019; Gallicchio & Scardapane, 2020), which is also the main strength of our method. However, due to the important influence the setup of a reservoir has on its performance, the hyperparameter search for a reservoir model is an exhaustive process subject to ongoing research (Trouvain et al., 2020; Mwamsojo et al., 2024). Our approach does not have a high number of tunable hyperparameters and does not require this additional optimization, thus it is perhaps a more suitable candidate for neural architecture search. Furthermore, with the connection to Koopman theory, our approach has a strong connection to dynamical systems.

## 3.2 Simple ODEs: Van der Pol Oscillator

We consider the Van der Pol oscillator system for a simple illustration of our method. This is a non-conservative oscillatory system with a nonlinear damping term, described as a two dimensional ODE

$$\begin{cases} \dot{h_1} &= h_2 \\ \dot{h_2} &= \mu(1 - h_1^2)h_2 - h_1, \end{cases}$$

where $\mu$ is a scalar parameter indicating the nonlinearity and the strength of the damping. In our experiment we set $\mu = 1$, so the dynamics are only mildly nonlinear. The training data points are created by solving an initial value problem for $t \in [0, 20]$ with $\Delta t = 0.1$ for 50 initial conditions, where each initial condition is chosen at random, $\boldsymbol{h}^0 \sim \text{Uniform}([-3, 3]^2)$. We use an explicit Runge-Kutta method of order 8 to solve the initial value problem. Validation and test data are generated similarly but for $t \in [0, 50]$, which is a longer timespan than the training data. We construct a KIRNN with a $tanh$ activation function and a single hidden layer of width 80. The model is then evaluated on test data; the averaged error and training time are reported in Table 1. One trajectory from the test set is visualized in Figure 2. It should be noted that predictions are made in an autoregressive manner, in other words, we start with an initial condition from the test dataset which is used to make the first prediction, and afterwards continue using this prediction as an input to predict the next state, without information from the ground-truth dataset. In the results we observe a very stable trajectory over a long prediction horizon, indicating stability of the model. This experiment is also significant due to the periodic nature of the system, which is captured with our model, although neural network architectures in general struggle to capture periodicity (Ziyin et al., 2020). Compared to the gradient-descent trained shPLRNN our method is much faster to train and achieves higher forecasting accuracy, with lower MSE as prediction error. Compared with an ESN, our model also has a shorter training time and a smaller error. Compared to both the shPLRNN and ESN methods, the hyperparameter search is simpler for our method since there are fewer hyperparameters to tune.

Our contribution rests on two fundamental ideas: data-aware sampling a hidden layer with SWIM, and using the Koopman operator for evolution in time, thus using two matrices (K and C) which could be replaced with one non-square matrix; thus we investigate the performance when one of these ideas is not employed. We consider four cases: 1) we sample from fixed distributions and do not use Koopman for time evolution, 2) we sample from fixed distributions and use Koopman, 3) we sample with SWIM and omit Koopman, and 4) use both SWIM and Koopman. Our findings are quantified in Table 2 and the results show that the combination of SWIM sampling and Koopman indeed outperforms naive approaches.

Table 2: Impact of using SWIM (data-aware) sampling, and using the Koopman operator compared to their exclusion. The results are obtained using the same Van der Pol problem setting explained previously.

| | SWIM | Koopman | MSE avg (min, max) |
|---|---|---|---|
| | X | X | 4.70e-2 (2.93e-2, 9.19e-2) |
| is used | X | ✓ | 1.46e-1 (2.14e-3, 6.82e-1) |
| | ✓ | X | 3.33e-2 (3.25e-2, 3.42e-2) |
| | ✓ | ✓ | 9.55e-4 (7.08e-4, 1.28e-3) |

## 3.3 Example with time delay embedding: Van der Pol Oscillator

For many real-world examples, it is not possible to observe the full state of a system. In this section, we use the same datasets as in the Van der Pol experiment (Section 3.2) but now only consider the first coordinate $h_1$ to be observable. We embed the data using a time-delay embedding of six followed by a principal component analysis (PCA) projection which reduces the dimensionality to two. A KIRNN with $tanh$ activation and a single hidden layer of width 80 is trained. Predicted trajectories from the initial test dataset state are shown in the bottom row of the right column of Figure 2. The fit time and MSE error are provided in Table 1, they are fairly similar to the fit time and error for the example where the full state is observed indicating that this model also captures the true dynamics, but now only requires a short time series of $h_1$ as an input.

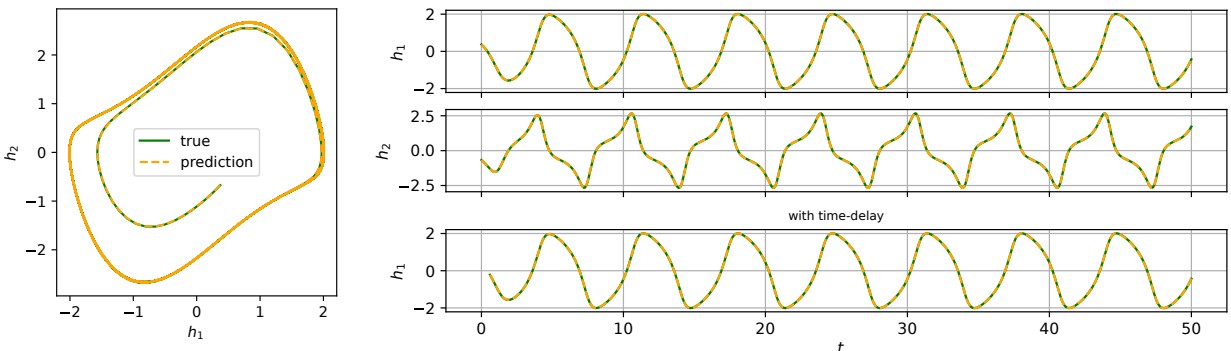

Figure 2: Comparison of true and predicted trajectories fror the Van der Pol experiments are shown for a test trajectory. Left: state space representation. Right: the top two rows show the full state system's first and second coordinate from Section 3.2, and the bottom most row shows the partially observed system from Section 3.3.

### 3.4 Examples of chaotic dynamics: Lorenz and Rössler systems

Chaotic systems pose a challenging forecasting problem from the class of dynamical systems. As an example, we consider the well-known Lorenz and Rössler systems in the chaotic regime.

The Lorenz-63 system (Lorenz, 1963) is defined as

$$\begin{cases} \dot{h_1} &= \sigma(h_2 - h_1) \\ \dot{h_2} &= h_1(\rho - h_3) - h_2 \ , \\ \dot{h_3} &= h_1 h_2 - \beta h_3, \end{cases}$$

where $\sigma, \rho, \beta$, are parameters that control the dynamics of the system. In our experiment, we set $\sigma = 10$, $\beta = \frac{8}{3}$, and $\rho = 28$, which means we are in the chaotic regime. Training data are generated by solving an initial value problem for $t \in [0, 5]$ with $\Delta t = 0.01$ for 50 initial conditions, where each initial condition is a random vector $\boldsymbol{h}^0 \sim \text{Uniform}([-20, 20] \times [-20, 20] \times [0, 50])$. We used an explicit Runge-Kutta solver of order 8. Validation and test data are generated similarly but for $t \in [0, 50]$. We normalize datasets to scale the values to the range $[-3, 3]$ to improve the training stability for the gradient-based method.

A KIRNN with a *tanh* activation and a single hidden layer of width 200 is trained. Naturally, due to the chaotic property of the system, agreement between predictions and numerical computations for long trajectories is not to be expected, since approximately similar states do not lead to approximately similar future states in chaos. However, predictions for a test trajectory, which is visualized in Figure 3, confirms that the model has learned the underlying attractor. Statistical estimates for this are reported in Table 1.

Furthermore, we consider the Rössler system (Rössler, 1976) given by

$$\begin{cases} \dot{h_1} &= -h_2 - h_3 \\ \dot{h_2} &= h_1 + \alpha h_2 \\ \dot{h_3} &= \beta + h_3(h_1 - \kappa), \end{cases}$$

where $\alpha, \beta, \kappa$, are parameters controlling the dynamics of the system. Here, we set $\alpha = 0.15$, $\beta = 0.2$, and $\kappa = 10$, which puts the system in the chaotic regime. The setup for data generation is similar to the Lorenz example; training data are generated by solving an initial value problem for $t \in [0, 10]$ with $\Delta t = 0.01$ with an 8th order explicit Runge-Kutta solver for 50 initial conditions, where each initial condition is random vector $\boldsymbol{h}^0 \sim \text{Uniform}([-20, 20] \times [-20, 20] \times [0, 40])$. Validation and test data are generated similarly but for $t \in [0, 200]$, an even longer prediction horizon. We again normalize datasets to the range $[-3, 3]$ to aid iterative training.

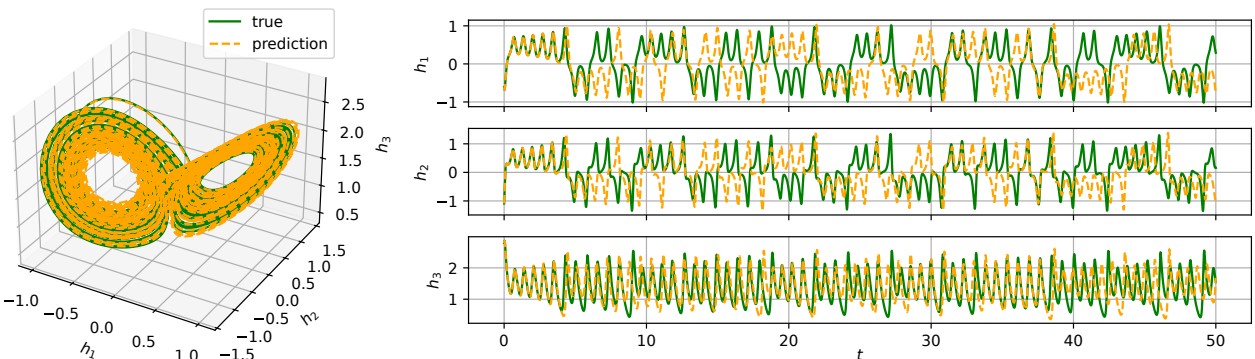

Figure 3: The results from the Lorenz experiment are shown for a test trajectory.
Left: state space representation of true and predicted trajectories. Right: trajectories obtained from the Lorenz model described in Section 3.4.

We use a KIRNN with 300 hidden layer nodes and *tanh* activation. A predicted test trajectory is shown in Figure 4, and we observe that the model again captures the underlying attractor well, similar to the Lorenz experiment.

The quantitative results for both chaotic systems are given in Table 1. Due to the chaotic nature of the two systems, an MSE evaluation is not suitable, and we therefore use an empirical KL divergence (EKL) between the points on the attractor in the test set and our approximation, detailed in Appendix C. We observe our model requires approximately half the fit time compared to the ESN and achieves comparable performance in terms of the EKL error. The good performance of ESN is not surprising for such common chaotic systems since good hyperparameters have been found as part of previous research efforts Viehweg et al. (2023). The results also suggest that our method for chaotic systems is significantly faster compared to the gradient-based model, as can be observed in Table 1, with comparable accuracy. An additional aspect is the lower effort spent on hyperparameter tuning for our method, as compared to both alternatives.

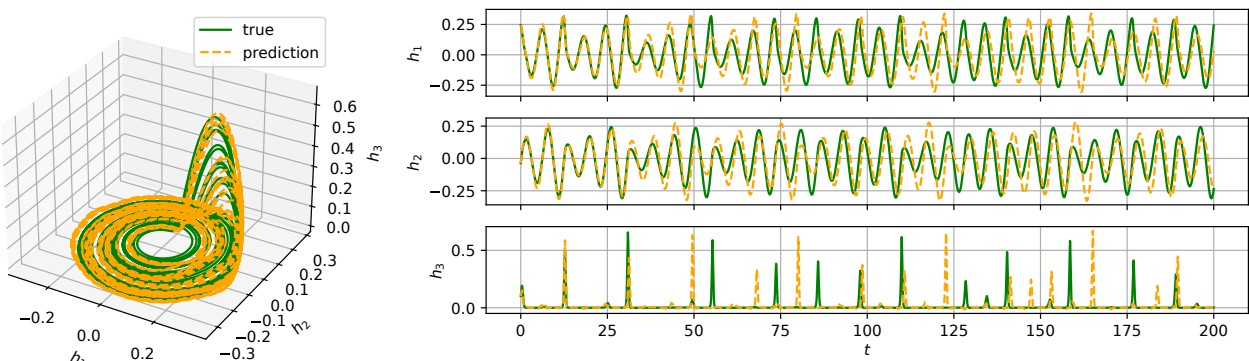

Figure 4: Trajectories from the Rössler experiment are shown for a test trajectory. Left: state space representation of true and predicted trajectories. Right: trajectories obtained from the Rössler model described in Section 3.4.

## 3.5 Interpretability via Koopman

We further explore the applicability of our method to a problem with unknown governing equations with real-world relevance and investigate interpretability in this context.

In recent work by Kern et al. (2025), the Koopman framework was used to obtain a predictive model for crowd dynamics and to extract underlying patterns of the dynamics. They find that a challenging nonlinear scenario is given by a periodic inflow of pedestrians into a corridor that narrows down and causes

a congestion at the narrowing point. This congestion is typically referred to as a *pedestrian bottleneck* in the crowd dynamics literature and is studied both in designed experiments (Liddle et al., 2009) and as a model validation benchmark (Kleinmeier et al., 2019). We follow the setup by Kern et al. (2025) and use the density-based representation of the crowd where each pedestrian is represented with a Gaussian kernel (as proposed by Seitz & Köster (2012)), and for the periodic inflow dataset we only consider the time period where the room is occupied with pedestrians, i.e. the period when the room is empty is cut out. Snapshots of the data (i.e., crowd density) are shown in Figure 5. Instead of using standard dictionary choices for EDMD, as done by Kern et al. (2025), we employ the sampled nodes as a dictionary; we use a hidden layer with 100 nodes and a ReLU activation.

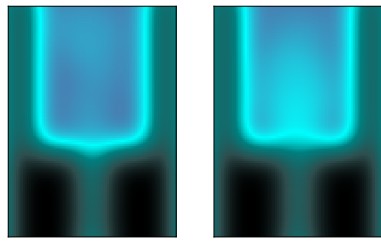

In this experiment we study the Koopman operator by visualizing the modes and eigenvalues in Figures 6a and 6b. The modes cover the scenario, with real and imaginary parts shown side by side. On the left the modes corresponding to the first eigenvalue are shown and their real part depicts the wall area of the domain, which remains unchanged throughout the simulation. The two other modes shown are for the second and fourth eigenvalue; the results for the third eigenvalue are ommited because they are very similar to the second eigenvalue. In these plots the dynamic part is seen which is likely related to the congestion and also picks up oscillatory patterns of the faster and slower moving parts of the crowd. Since we use numerical algorithms to approximate the Koopman operator, the discretization might lead to spurious eigenvalues, thus as a precaution we use an algorithm by Colbrook & Townsend (2024) to approximate the squared relative residual of eigenvalues and depict the residuals with a coloring scheme in Figure 6a. For our model, the eigenvalues are all inside the unit circle, and some of them are associated with a high residual (the dark red points). Due to their small magnitude, they are not impacting the predictions significantly. As all eigenvalues have a magnitude smaller than 1 the predictions will remain stable.

Figure 5: Crowd density for a pedestrian bottleneck scenario. Left: no congestion. Right: congestion appears at corridor narrowing. Walls are distinguished by the dark/black regions.

We emphasize that an analysis like this is only possible because of the direct connection of our RNN to Koopman theory. The predictions of our sampled network are shown in Figure 6c for two locations over time. The predictions are a smoother curve compared to the true density value and slight inaccuracies can be seen. The average fit time was 1.28 seconds, and evaluated on a test set the average MSE was $2.10e-4$ (max: $3.67e-4$, min: $7.56e-05$).

### 3.6 Example with control inputs: Controlled Van der Pol Oscillator

We consider again the Van der Pol oscillator, where now the second coordinate, $h_2$, is controlled with an external input $x$, and using a KIRNN model we perform nonlinear control as in Algorithm 3.

The data is obtained for $t \in [0, 50\Delta t]$ with $\Delta t = 0.05$, with 150 initial conditions, where $\boldsymbol{h}^0 \sim$ Uniform($[-3,3]^2$) and $x^0 \sim$ Uniform($[-3,3]$). To integrate the true system in time, we use an explicit Runge-Kutta method of order 5(4). The control input data $x$ in the training set are not obtained from a controller with a particular target state, but instead, random values of $x$ are applied to the trajectories over time.

The KIRNN consists of a hidden layer of width 32 for the control inputs (denoted as $\mathcal{G}_{\hat{M}}$) and a hidden layer of width 128 for the state (denoted as $\mathcal{F}_M$) and $tanh$ activation. This network is then passed as a surrogate model to a LQR. Using the system matrix $K$ and control input matrix $B$ to solve an optimization problem, the LQR can successfully steer the state to the target state (see Figure 7). This experiment highlights a key advantage of our model, which allows for modeling a nonlinear system such as the Van der Pol oscillator using a linear controller such as LQR, due to our definition of linear maps $K$ and $B$. This implies that the well-established tools from linear control theory can be applied to non-linear systems using our method.

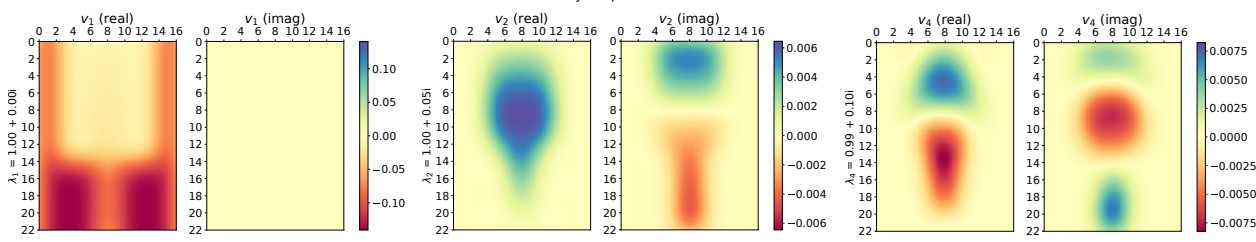

(a) Koopman modes corresponding to the first (left), second (middle) and fourth (right) eigenvalues sorted by magnitude. The real and imaginary parts are shown separately side-by-side.

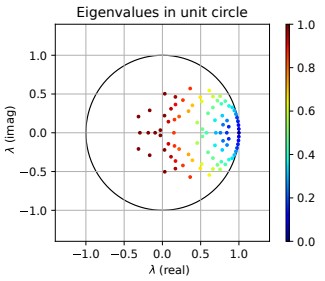

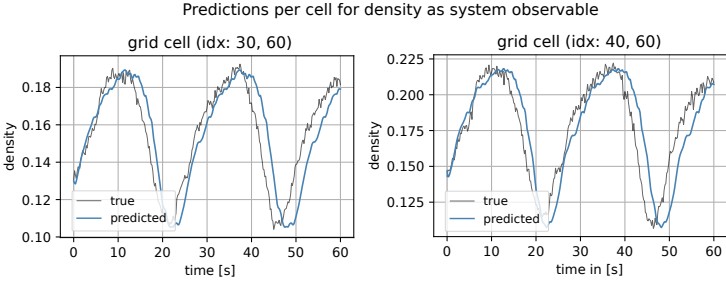

(b) Eigenvalues of the Koopman operator in relation to the unit circle. The colormap shows the estimated residuals calculated according to Algorithm 3 by Colbrook & Townsend (2024).

(c) Comparison of the true (blue) and predicted (black) density at selected grid points. The left plot shows the density evolution for a point before the narrowing of the corridor which is slightly offset to the left. The right plot shows the density before the narrowing of the corridor and it is centered.

Figure 6: Results from the crowd density model are shown as operator properties (a, b) and predictive ability (c).

We consider five different runs, where only the random seed is varied, and obtain the mean controller cost to be 13.96 (min: 5.43, max: 22.99) and the mean training time of 0.5699 seconds. The norm of the state is also tracked over time, for five different runs we show the norms and the pointwise mean (over the runs) in Figure 7. The authors are not aware of applications of state-of-the-art recurrent networks for nonlinear control using LQR, thus there is no comparison. Our goal for this experiment was to show how KIRNNs are compatible with tools from linear control theory, in particular LQR. Further work is required to evaluate and compare this method to nonlinear control tools. In addition, choices such as how long one predicts before applying the control to the dynamical system is something we have looked into with different lengths, but more is left to understand its limitations.

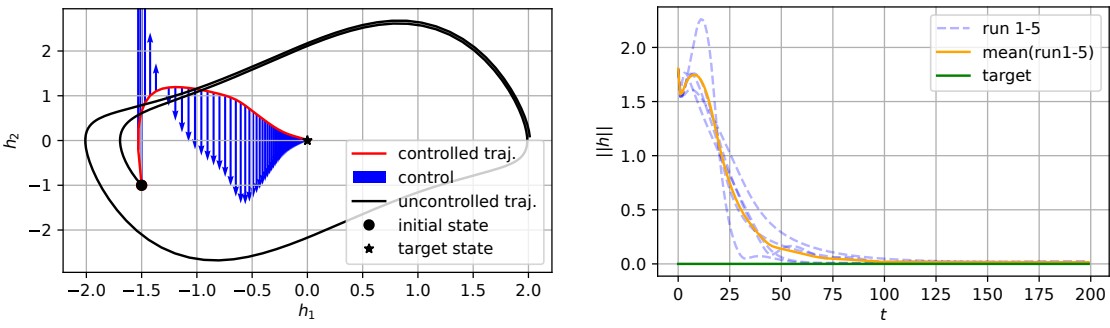

Figure 7: Controlled (i.e. forced) Van der Pol experiment (Section 3.6) for initial condition $\boldsymbol{h}_0 = [-1.5, -1]^\mathsf{T}$. Left: state space representation of controlled and uncontrolled trajectories. Right: $L^2$ norm of the controlled trajectory for five different runs and the $L^2$ norm of the target state.

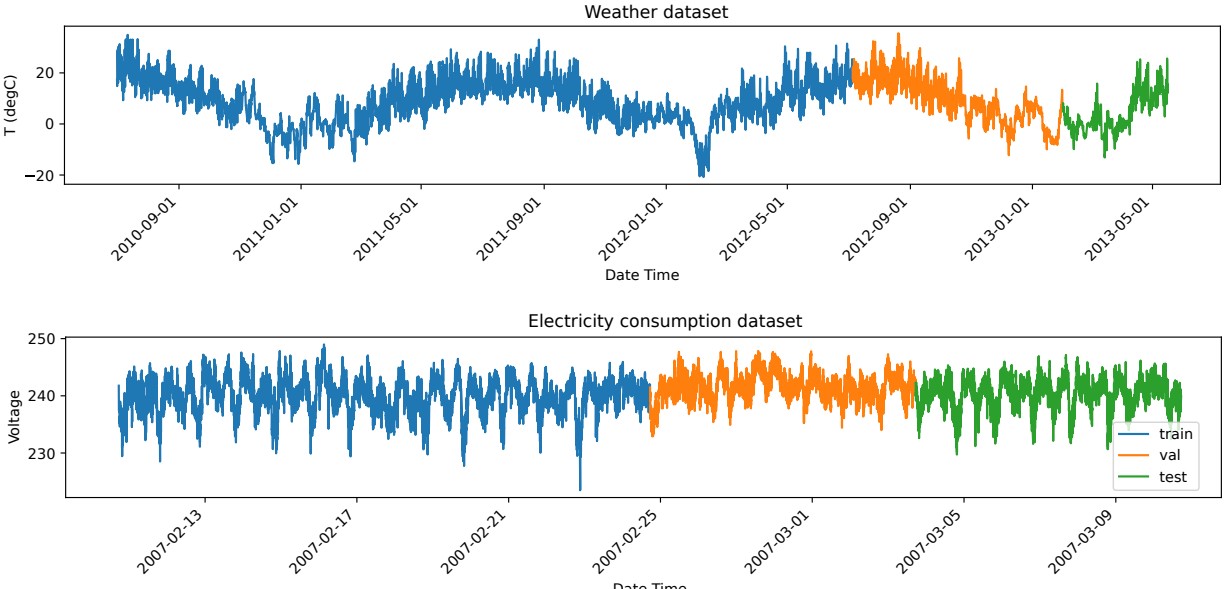

Figure 8: The two datasets for real-world experiments colored according to the data splits: train (blue), validation (orange), and test (green).

### 3.7 Examples with real-world data

Finally, we will end this section with two examples using real-world data, namely weather data and individual power consumption.

#### 3.7.1 Weather data

We apply our approach to model the climate data presented in TensorFlow (2024). The dataset (for Biogeochemistry, 2024) contains a time series of 14 weather parameters recorded in Jena (Germany) between January 1st, 2009, and December 31st, 2016. The data contains inconsistent/missing date and time values, leading to gaps and overlaps between measurements. We extracted the longest consecutive time period and thus worked with the data between July 1st, 2010, and May 16th, 2013. We additionally downsampled the time series from the original 10-minute to 1-hour measurements. Then, the first 70% of records were used as the train set, the next 20% as the validation, and the remaining 10% as the test set. Identically to TensorFlow (2024), we pre-processed the features and added `sin` and `cos` time-embeddings of hour, day, and month. We plot the dataset, indicating the train-validation-test split with colors in Figure 8. The data offers freedom in choosing the sizes of the time delay and prediction horizons. We decided to fix the time delay to one week and set two separate experiments with a prediction horizon of one day and one week. In the KIRNN and ESN experiments, we performed a grid search for each model with hyperparameters specified in Table 4. Table 1 shows the averaged training time and error metrics for the two selected horizons (day and week). We observe that the models perform similarly in the case of a shorter horizon, while sampling offers much faster training compared to a gradient-based method. Compared to an ESN the training time of a KIRNN is a bit longer, but the search for hyperparameters was shorter for a KIRNN. When considering a one-week horizon, shPLRNN outperforms, while KIRNN is still orders of magnitude faster. We also note the prediction horizon does not influence the training time of the sampled model that agrees with the Algorithm 2. When comparing predictions for the longer horizon in Figure 9, we notice that the ESN struggles to predict the high-frequency fluctuations of the measurement, but the KIRNN and shPLRNN successfully capture them. Figure 9 also highlights the deficiency of the MSE metric because a low mean error does not always correspond to accurate predictions, as illustrated by all models. Overall, we conclude that KIRNNs can successfully capture chaotic real-world dynamics and produce results comparable to the iterative models while offering a significant speed-up in training.

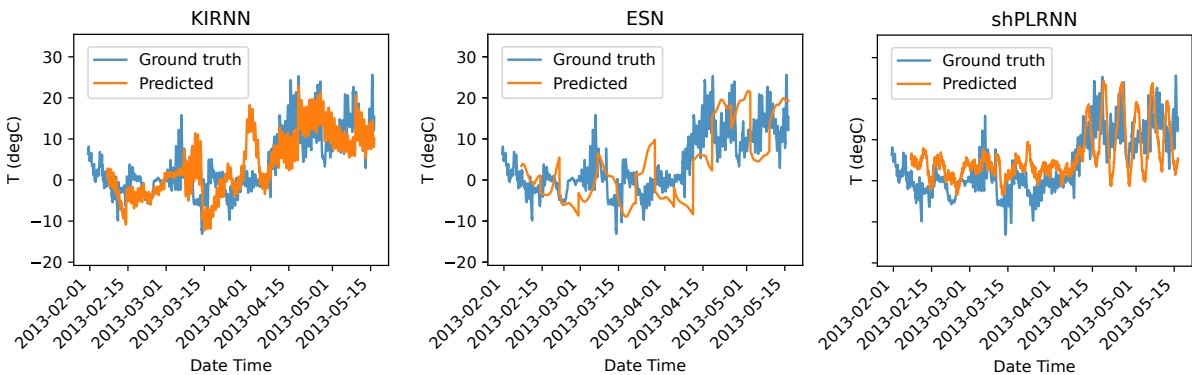

Figure 9: The predictions of the best models on the test dataset for the horizon of one week: KIRNN (left), ESN (middle) and shPLRNN (right).

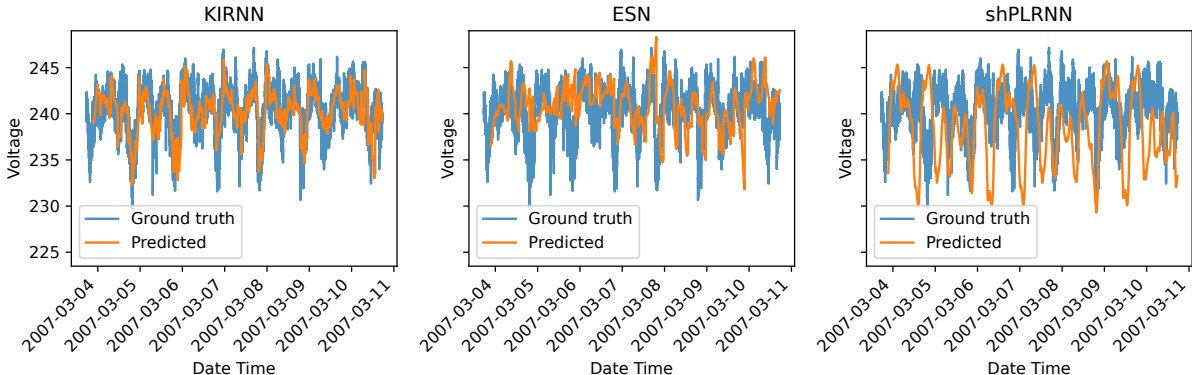

Figure 10: Test set predictions of individual household electricity consumption in voltage by: KIRNN, (left) ESN (middle) and shPLRNN (right).

### 3.7.2 Electricity consumption

We use the individual power consumption dataset by Hebrail & Berard (2006) and predict the voltage feature. We take an approach similar to the one for weather predictions. This dataset contains measurements made in a one-minute interval, and we consider a period of four weeks as our dataset. Two weeks are used as training data, one week as validation and one as test data, these are ordered sequentially in time, similar to the split of the weather dataset. We also here add `sin` and `cos` time-embeddings of hour and day as features. In the experiments performed we use a time-delay window of 240 consecutive time steps, which is 4 hours, as input. We then predict a horizon of 120 steps (so 120 minutes or 2 hours). The KIRNN has 64 hidden nodes and a *tanh* activation. The results are plotted in Figure 10. Our results are reported in Table 1 and Figure 10; we notice a good performance for the KIRNN both in terms of accuracy and training time, while the ESN has worse accuracy. The shPLRNN required extensive hyperparameter tuning and a notably higher hidden dimension to achieve acceptable performance, and it still has an overall worse performance for this task.

## 4 Conclusion

We introduce an efficient and interpretable training method for recurrent neural networks by combining ideas from random feature networks and Koopman operator theory. Our method circumvents the common problems associated with conventional RNNs, such as EVGP, and presents a computationally efficient alternative that performs comparatively in terms of accuracy. This is done by sampling the hidden layers, and solving for the final linear layers with extended dynamic mode decomposition (EDMD). In addition, the connection

we make to the Koopman operator allows us to apply tools from linear control theory to RNNs. We also use this connection to analyze the dynamics learned by studying the Koopman modes and the spectrum.

The training method we use involves the solution of a large, linear system. The complexity of solving this system depends cubically on the minimum number of neurons and the number of data points (respectively, time steps). This means if both the network and the number of data points are large, the computational time and memory demands for training may present challenges. The requirement of being able to express states of the system limits the approach. In addition, we observe that the training method does not perform well for intrinsically high dimensional data. This combined means that extending our approach to tasks such as natural language processing and computer vision remains out of reach without modifications in architecture and training.

Remaining challenges include extending the theory shown in this paper to controlled systems. We also wish to extend the theoretical results to include the sampling scheme. Bridging Koopman theory for continuous dynamical systems with NeuralODEs is also an interesting avenue for future research. For the controlled setting, we only demonstrated the use of linear control applied to KIRNN on a simple example enabling nonlinear control, but further experiments need to be done in order to compare this approach against nonlinear control methods. Lazar (2025) proposes applying the Koopman operator to controlled systems differently from previous literature, and there are clear connections to our work, as we also lift the control input. Investigating these connections further would also be of interest.

## Acknowledgments

E.B., I.B., and F.D. are thankful to the Deutsche Forschungsgemeinschaft (DFG, German Research Foundation) - for funding through project no. 468830823, and also acknowledge the association to DFG-SPP-229. F.D. and A.C. are supported by the TUM Georg Nemetschek Institute - Artificial Intelligence for the Built World. Z.M. is grateful to the Bundesministerium für Bildung und Forschung (BMBF, Federal Ministry of Education and Research) for funding through project OIDLITDSM, No. 01IS24061. We also thank Lukas Eisenmann and Florian Hess for their helpful suggestions and guidance in training the shPLRNNs used in this paper. Furtheremore, we are grateful to Daniel Lehmberg for developing and maintaining the `datafold` library which was useful for developing the codebase and the sampled RNN experiments. Finally, we are indebted to Sabrina Kern and Gerta Köster for sharing their codebase and data on the interepretability in pedestrian dynamics with Koopman.

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

## Appendix

## A   Koopman operator and extended dynamic mode decomposition

In this section we give a more thorough introduction to the Koopman operator and its use in dynamical system theory. In addition, we also explain extended dynamic mode decomposition (EDMD), which is used for the finite-dimensional approximation of the Koopman operator we use in the main paper. Consider a dynamical system $(\mathcal{H}, F)$, where the state space $\mathcal{H}$ can be any topological space, and $F \colon \mathcal{H} \to \mathcal{H}$ is a flow map of a dynamical system. We impose more structure on our system by requiring our state space $\mathcal{H}$ to be a (finite or infinite-dimensional) Hilbert space with suitable measure $\mu$, sigma algebra $\Sigma$, and require $F$ to be $\Sigma$-measurable. We start by considering $(\mathcal{H}, F)$ to be a discrete-time system, which are the systems we mainly work with in this paper. We can then write the evolution as

$$\boldsymbol{h}_{t+1} \,=\, F(\boldsymbol{h}_t), \quad \boldsymbol{h}_t \in \mathcal{H} \subseteq \mathbb{R}^{d_h}, \quad t \in \mathbb{N}_{\geq 0}. \tag{15}$$

The analysis of the evolution in the original state space can be difficult, especially when $F$ is non-linear. Using Koopman theory we can take a different approach, and instead consider the evolution of observables $\phi \colon \mathcal{H} \to \mathbb{C}$ instead of the states themselves. The evolution of the observables is then captured by the *Koopman operator* $\mathcal{K}$,

$$[\mathcal{K}\,\phi](\boldsymbol{h}) := (\phi \circ F)(\boldsymbol{h}).$$

As long as the space of observables is a vector space, the Koopman operator is linear and we may analyse the dynamical system with non-linear $F$ using spectral analysis, with the caveat that in the majority of cases the domain of $\mathcal{K}$ is infinite dimensional. The choice of this domain, which we call $\mathcal{F}$ here, is crucial, as $\phi \circ F$ must belong to $\mathcal{F}$ for all $\phi \in \mathcal{F}$. Assuming $F$ is measure-preserving—which is common in ergodic theory—one can address the issue by setting

$$\mathcal{F} = L^2(\mathcal{H}, \mu_h) := \left\{ \phi \colon \mathcal{H} \to \mathbb{F} \;\middle|\; \|\phi\|_{L^2(\mathcal{H}, \mu_h)} = \left( \int_{\mathcal{H}} |\phi(\boldsymbol{h})|^2 \, \mu_h(d\boldsymbol{h}) \right)^{\frac{1}{2}} < \infty \right\}. \tag{16}$$

where $\mathbb{F} = \mathbb{R}$ or $\mathbb{C}$. As we consider spectral analysis in this section, we let $\mathbb{F} = \mathbb{C}$. If $F$ is measure-preserving, then $\mathcal{K}$ is an isometry and the issue is resolved. The map $\mathcal{K}$ might still not be well-defined, as $\phi_1, \phi_2 \in L^2(\mathcal{H}, \mu_h)$ may differ only on a null set, yet their images under $\mathcal{K}$, could differ over a set of positive measure. To exclude this possibility, $F$ must be $\mu_h$-nonsingular, meaning that for each $H \subseteq \mathcal{H}$, $\mu_h(F^{-1}(H)) = 0$ if $\mu_h(H) = 0$. Once the function space $\mathcal{F}$ is chosen, making sure that $\mathcal{K}$ is well-defined, we may apply spectral analysis. A Koopman eigenfunction $\varphi_k \in \mathcal{F}$ corresponding to a Koopman eigenvalue $\lambda_k \in \sigma(\mathcal{K})$ satisfies

$$\varphi_k(\boldsymbol{h}_{t+1}) \,=\, \mathcal{K}\varphi_k(\boldsymbol{h}_t) \,=\, \lambda_k \, \varphi_k(\boldsymbol{h}_t).$$

When the state space $\mathcal{H}$ is a subset of $\mathbb{R}^{d_h}$, under certain assumptions on the space of eigenfunctions, we can evolve $\boldsymbol{h}$ using the spectrum of $\mathcal{K}$. More concretely, let $\Phi \colon \mathcal{H} \to \mathbb{C}^{d_h}$ be a vector of observables, where each observable $\phi_i(\boldsymbol{h}) = \boldsymbol{h}_i$, and $\boldsymbol{h}_i$ is the $i$th component of $\boldsymbol{h}$. Assuming $\phi_i \in \text{Span}\{\varphi_k\} \subset \mathcal{F}$, we can write

$$\phi_i(\boldsymbol{h}) = \sum_k c_k^{\phi_i} \varphi_k.$$

Here, the coefficients $c_k^{\phi_i} \in \mathbb{C}$ are often called *Koopman modes* associated with the observable $\phi_i$. We then have

$$\mathcal{K}\,\phi_i \,=\, \mathcal{K}\sum_k c_k^{\phi_i}\,\varphi_k \,=\, \sum_k c_k^{\phi_i}\,\mathcal{K}\,\varphi_k \,=\, \sum_k c_k^{\phi_i}\,\lambda_k\,\varphi_k,$$

because the operator $\mathcal{K}$ is linear. Iterative application of $\mathcal{K}$ a number of $t \in \mathbb{N}$ times yields

$$\boldsymbol{h}_t = \Phi(\boldsymbol{h}_t) = (\underbrace{[\mathcal{K} \circ \cdots \circ \mathcal{K}]}_{t}\,\phi)(\boldsymbol{h}_0) = \sum_k \lambda_k^t \; \varphi_{\lambda_k}(\boldsymbol{h}_0) \; \boldsymbol{c}_k^{\Phi}. \tag{17}$$

This process is known as *Koopman mode decomposition* (KMD). This reveals one of the true strengths of the Koopman theory: understanding the stability of a system through analyzing the spectrum. Up until now we have considered a discrete dynamical system, but the Koopman theory can also be extended to continuous systems where $\boldsymbol{h}$ is a function of continuous time $t$ and its evolution is given by

$$\dot{\boldsymbol{h}}(t) = v(\boldsymbol{h}(t)), \quad \boldsymbol{h}(t) \in \mathcal{H}, t \in \mathbb{R}_{\geq 0},$$

with a vector field $v$ such that integrating $\dot{h}$ for time $t$ results in the flow $F^t$. For any $t$, the flow map operator denoted by $F^t \colon \mathcal{H} \to \mathcal{H}$ is defined as

$$\boldsymbol{h}(t) = F^t(\boldsymbol{h}) = \boldsymbol{h}(0) + \int_0^t v(\boldsymbol{h}(\tau))d\tau,$$

which maps from an initial condition $\boldsymbol{h}(0)$ to point on the trajectory at time $t$. We can then define the Koopman operator for each $t \in \mathbb{R}_{\geq 0}$ ,

$$[\mathcal{K}^t \phi](\boldsymbol{h}) = (\phi \circ F^t)(\boldsymbol{h}),$$

where $\phi \in \mathcal{F}$. The set of all these operators $\{\mathcal{K}^t\}_{t\in\mathbb{R}_{\geq 0}}$ forms a semigroup with an infinitesimal generator $\mathcal{L}$. With some assumption on the continuity of the semigroup, the generator is the Lie derivative of $\phi$ along the vector field $v(\boldsymbol{h})$ and can be written as

$$[\mathcal{L} \phi](\boldsymbol{h}) = \lim_{t \downarrow 0} \frac{[\mathcal{K}^t \phi](\boldsymbol{h}) - \phi(\boldsymbol{h})}{t} = \frac{d}{dt}\phi(\boldsymbol{h}(t))\Big|_{t=0} = \nabla\phi \cdot \dot{\boldsymbol{h}}(0) = \nabla\phi \cdot v(\boldsymbol{h}(0)).$$

The eigenfunction and eigenvalue are in the continuous time case scalars and functions satisfying

$$[\mathcal{K}^t \varphi_k](\boldsymbol{h}) = e^{\lambda_k t}\, \varphi_k(\boldsymbol{h}),$$

where $\{e^{\lambda_k}\}$ are the eigenvalues of the operator $\mathcal{K}^t$, and $\lambda_k$ are eigenvalues of the generator $\mathcal{L}$. This allows us to use the Koopman theory for continuous dynamical systems as well, and possibly make the connection for NeuralODEs and Koopman theory in similar fashion we have done with discrete system Koopman operator and RNN.

As the Koopman operator is infinite dimensional it is not possible to apply it directly, which raises the need for a method to create a finite approximation of $\mathcal{K}$ and its spectrum, namely the extended dynamic mode decomposition.

## A.1 Extended dynamic mode decomposition

As we are mostly working with discrete systems in this paper, we focus on approximating the Koopman operator $\mathcal{K}$ for discrete dynamical systems. The way to approximate $\mathcal{K}$ is by extended dynamic mode decomposition (EDMD), which is an algorithm that provides a data driven finite dimensional approximation of the Koopman operator $\mathcal{K}$ through a linear map $K$. The spectral properties of $K$ subsequently serve to approximate those of $\mathcal{K}$. Utilizing this approach enables us to derive the Koopman eigenvalues, eigenfunctions, and modes. Here, we provide a brief overview of EDMD. For further details, refer to Williams et al. (2015). The core concept of EDMD involves approximating the operator's action on $\mathcal{F} = L^2(\mathcal{H}, \mu_h)$ by selecting a finite dimensional subspace $\widetilde{\Psi}_M \subset \mathcal{F}$. To define this subspace, we start by choosing a dictionary $\Psi_M = \{\psi_i : \mathcal{H} \to \mathbb{R} \mid i = 1, \ldots, M\}$. We then have

$$\Psi_M(\boldsymbol{h}) = [\psi_1(\boldsymbol{h}), \psi_2(\boldsymbol{h}), \ldots, \psi_M(\boldsymbol{h})]^\mathsf{T} \in \mathbb{R}^{d_h},$$

and we let the finite dimensional subspace $\widetilde{\Psi}_M$ be

$$\widetilde{\Psi}_M = \mathrm{Span}\{\psi_1, \psi_2, \cdots, \psi_M\} = \{\boldsymbol{a}^\mathsf{T} \Psi_M : \boldsymbol{a} \in \mathbb{C}^M\} \subset \mathcal{F}.$$

The action of the Koopman operator on $\phi \in \widetilde{\mathcal{F}}_M$ due to linearity is

$$\mathcal{K}\phi = \boldsymbol{a}^\mathsf{T} \mathcal{K}\Psi_M = \boldsymbol{a}^\mathsf{T} \Psi_M \circ F.$$

Assuming that the subspace $\widetilde{\Psi}_M$ is invariant under $\mathcal{K}$, i.e., $\mathcal{K}(\widetilde{\Psi}_M) \subseteq \widetilde{\Psi}_M$, we can write $\mathcal{K}\phi = \boldsymbol{b}^\mathsf{T}\mathcal{F}_M$ for any $\phi \in \widetilde{\Psi}_M$. It follows that $\mathcal{K}|_{\widetilde{\Psi}_M}$ is finite dimensional and can be written as a matrix $K \in \mathbb{R}^{M \times M}$ such that $\boldsymbol{b}^\mathsf{T} = \boldsymbol{a}^\mathsf{T}K$. This means we can use a finite approximation to compute the action of the Koopman operator on $\phi \in \widetilde{\mathcal{F}}_M$, i.e.,

$$K\phi = \boldsymbol{a}^\mathsf{T}K\Psi_M = \boldsymbol{b}^\mathsf{T}\Psi_M = \mathcal{K}\phi.$$

When $\widetilde{\Psi}_M$ is not an invariant subspace under the Koopman operator $\mathcal{K}$, $K$ becomes an approximation. Decomposing $\mathcal{K}\phi = \boldsymbol{b}^\mathsf{T}\Psi_M + \rho$, where $\rho \in L^2(\mathcal{H}, \mu_h)$, EDMD approximates $\mathcal{K}$ by using data

$$H = [\boldsymbol{h}_1, \boldsymbol{h}_2, \dots, \boldsymbol{h}_N] \in \mathbb{R}^{d_h \times N}, \quad H' = [\boldsymbol{h}'_1, \boldsymbol{h}'_2, \dots, \boldsymbol{h}'_N] \in \mathbb{R}^{d_h \times N},$$

where $\boldsymbol{h}'_n = F(\boldsymbol{h}_n)$. Then, to find $K$, the algorithm EDMD minimizes the following cost function

$$\mathcal{J} = \frac{1}{2}\sum_{n=1}^{N} \|\rho(\boldsymbol{h}_n)\|^2 = \frac{1}{2}\sum_{n=1}^{N} \left\|\boldsymbol{a}^\mathsf{T}\left(\Psi_M(\boldsymbol{h}'_n) - K\Psi_M(\boldsymbol{h}_n)\right)\right\|^2. \tag{18}$$

Setting

$$\Psi_M(H) = [\Psi_M(\boldsymbol{h}_1), \Psi_M(\boldsymbol{h}_2), \dots, \Psi_M(\boldsymbol{h}_N)] \in \mathbb{C}^{M \times N}$$

and equivalently for $\Psi_M(H')$, a solution to Equation (18) is given by

$$K = \Psi_M(H')\,\Psi_M(H)^+ \tag{19}$$

where $\Psi_M(H)^+$ denotes the pseudo-inverse. Upon obtaining $K$, we find approximations of eigenfunctions

$$\varphi_k = \xi_k \Psi_M,$$

where $\xi_k$ is a left eigenvector of $K$. Finally, denoting $\boldsymbol{\varphi}\colon \mathcal{H} \to \mathbb{C}^M$ as the function $\boldsymbol{h} \mapsto [\varphi_1(\boldsymbol{h}) \oplus \varphi_2(\boldsymbol{h}) \oplus \cdots \oplus \varphi_K(\boldsymbol{h})]$, we approximate the Koopman modes by

$$C = \operatorname*{arg\,min}_{\tilde{C} \in \mathbb{C}^{d_y \times M}} \|\Phi(H) - \tilde{C}\boldsymbol{\varphi}(H)\|_{Fr}^2 = \operatorname*{arg\,min}_{\tilde{C} \in \mathbb{C}^{d_y \times M}} \|H - \tilde{C}\boldsymbol{\varphi}(H)\|_{Fr}^2,$$

where $\|\cdot\|_{Fr}$ is the Frobenius norm and $\Phi(\boldsymbol{h}) = \boldsymbol{h}$ is the identity function. We may now approximate Equation (17) as

$$\boldsymbol{h}_t = \Phi(\boldsymbol{h}_t) = C\,\Lambda^t\boldsymbol{\varphi}(\boldsymbol{h}_0),$$

where $\Lambda$ is a diagonal matrix with the corresponding eigenvalues.

In many applications, one is not necessarily interested in the spectral analysis, but only prediction. One then typically sets $\mathbb{F} = \mathbb{R}$, solves for $K$ in an exact way, but solves for $C$ as

$$C = \operatorname*{arg\,min}_{\tilde{C} \in \mathbb{R}^{d \times M}} \|\Phi(H) - \tilde{C}\Psi_M(H)\|_{Fr}^2 = \operatorname*{arg\,min}_{\tilde{C}} \|H - \tilde{C}\Psi_M(H)\|_{Fr}^2.$$

For prediction one simply applies $K$ several times,

$$\boldsymbol{h}_t = CK^t\Psi_M(\boldsymbol{h}_0).$$

Due to computational stability, one usually predicts $\boldsymbol{h}_t$ step by step, that is, maps it down to the state space and maps back to the observable image space, before applying $K$ again, instead of simply applying $K^t$. It is also worth noting that one may be more interested in mapping to an output of a function $f \in \mathcal{F}$ instead, and then one simply swaps $\Phi$ with $f$ when approximating $C$.

To conclude this introduction to EDMD, we do want to mention that the set of eigenfunctions we find through the EDMD algorithm comes with its own set of issues, such as spectral pollution, and efforts to mitigate certain issues has spawned extensions to EDMD, e.g., measure-preserving EDMD and residual DMD (Colbrook, 2023; Colbrook et al., 2023).

### A.2 Controlled dynamical systems and the Koopman operator

Extending Koopman theory to controlled systems can be done in several ways, and we opt to follow Korda & Mezić (2018a) and limit ourselves to linear controlled systems,

$$\boldsymbol{h}_t = F(\boldsymbol{h}_{t-1}, \boldsymbol{x}_t) = A_h \boldsymbol{h}_{t-1} + A_x \boldsymbol{x}_t, \quad \boldsymbol{y}_t = A_y \boldsymbol{h}_t. \tag{20}$$

Rewriting the input and the evolution operator slightly allows us to apply the Koopman operator and its theory as described in previous sections. Let

$$\tilde{\boldsymbol{h}} = \boldsymbol{h} \oplus \tilde{\boldsymbol{x}}$$

where $\boldsymbol{h} \in \mathcal{H}$ and $\tilde{\boldsymbol{x}} \in \ell(\mathcal{X})$ are concatenated, with $\ell(\mathcal{X})$ is the space of all countable sequences $\{\boldsymbol{x}_i\}_{i=1}^{\infty}$ such that $\boldsymbol{x}_i \in \mathcal{X}$. Then we can rewrite the evolution operator $F \colon \mathcal{H} \times \mathcal{X} \to \mathcal{H}$ to $\tilde{F} \colon \mathcal{H} \times \ell(\mathcal{X}) \to \mathcal{H}$, where

$$\tilde{\boldsymbol{h}}_t = \tilde{F}(\tilde{\boldsymbol{h}}_{t-1}) = [F(\boldsymbol{h}_{t-1}, \tilde{\boldsymbol{x}}(0))] \oplus [\mathcal{S}\tilde{\boldsymbol{x}}]$$

where $\tilde{\boldsymbol{x}}(i) = \boldsymbol{x}_i \in \tilde{\boldsymbol{x}}$ and $\mathcal{S}$ is the left-shift operator, i.e., $\mathcal{S}(\tilde{\boldsymbol{x}}(i)) = \tilde{\boldsymbol{x}}(i+1)$. The Koopman operator $\mathcal{K}$ can be applied to $\tilde{F}$ with observables $\phi \colon \mathcal{H} \times \ell(\mathcal{X}) \to \mathbb{C}$, and the rest of the Koopman theory follows.

When approximating the Koopman operator for controlled systems with EDMD, the dictionary we choose needs alteration due to the domain $\mathcal{H} \times \ell(\mathcal{X})$ being infinite dimensional. Korda & Mezić (2018a) propose dictionaries that are both computable and enforce the linearity relationship assumed in Equation (20). The dictionaries to be considered are of the form

$$\psi_i(\boldsymbol{h}, \tilde{\boldsymbol{x}}) = \psi_i^{(h)}(\boldsymbol{h}) + \psi_i^{(x)}(\tilde{\boldsymbol{x}}),$$

where $\psi_i^{(x)} \colon \ell(\mathcal{X}) \to \mathbb{R}$ is a linear functional and $\psi_i^{(h)} \in \mathcal{F}$. The new dictionaries can be written as

$$\Psi_M^{(h)} = \{\psi_1^{(h)}, \ldots, \psi_M^{(h)}\}, \quad \Psi_{\tilde{M}}^{(x)} = \{\psi_1^{(x)}, \ldots, \psi_{\tilde{M}}^{(x)}\}.$$

Note that the number of observables $M$ and $\tilde{M}$ can differ, even though in the main paper we typically set $\tilde{M} = M$. If they differ, the matrix $B$ will map from $\mathbb{F}^{\tilde{M}}$ to $\mathbb{F}^M$. Once the dictionaries are set, we simply solve the optimization problem

$$\underset{K \in \mathbb{F}^{M \times M}, B \in \mathbb{F}^{M \times \tilde{M}}}{\arg\min} \frac{1}{2} \sum_{n=1}^{N} \|\Psi_M^{(h)}(\boldsymbol{h}_n') - (K \Psi_M^{(h)}(\boldsymbol{h}_n) + B \Psi_{\tilde{M}}^{(x)}(\boldsymbol{h}_n))\|^2$$

with the analytical solution being

$$[K, B] = \Psi_M^{(h)}(H')(\Psi_M^{(h)}(H) \oplus \Psi_{\tilde{M}}^{(x)}(H))^+,$$

where $\Psi_M^{(h)}(H) \oplus \Psi_{\tilde{M}}^{(x)}(H) \in \mathbb{R}^{(M+\tilde{M}) \times N}$ is the concatenation of the two matrices. Approximating $C$ is done in the same manner as for uncontrolled systems, as it only needs to learn how to map from $\Psi_M^{(h)}(\mathcal{H})$ to $\mathcal{H}$. For further details, see Korda & Mezić (2018a).

## B Theory

In this section we give the necessary assumptions and proofs for the theoretical results in the main paper. We start by defining the state space $\mathcal{H}$ and input space $\mathcal{X}$, following the setup from Bolager et al. (2023). We set

$$d_{\mathbb{R}^{d_z}}(\boldsymbol{z}, A) = \inf\{d(\boldsymbol{z}, \boldsymbol{a}) \colon \boldsymbol{a} \in A\},$$

where $d$ is the canonical Euclidean distance in the space $\mathbb{R}^{d_z}$. The medial axis is defined as

$$\mathrm{Med}(A) = \{\boldsymbol{h} \in \mathbb{R}^{d_z} \colon \exists \boldsymbol{p} \neq \boldsymbol{q} \in A, \|\boldsymbol{p} - \boldsymbol{z}\| = \|\boldsymbol{q} - \boldsymbol{z}\| = d_{\mathbb{R}^{d_z}}(\boldsymbol{z}, A)\}$$

and the reach is the scalar

$$\tau_A = \inf_{\boldsymbol{a} \in A} d_{\mathbb{R}^{d_z}}(\boldsymbol{a}, \mathrm{Med}(A)),$$

i.e., the point in $A$ that is closest to the projection of points in $A^c$.

**Definition 2.** *Let $\widetilde{\mathcal{H}}$ be a nonempty compact subset of $\mathbb{R}^{d_h}$ with reach $\tau_{\widetilde{\mathcal{H}}} > 0$ and equivalently for $\widetilde{\mathcal{X}} \in \mathbb{R}^{d_x}$. The input space $\mathcal{H}$ is defined as*

$$\mathcal{H} = \{\boldsymbol{h} \in \mathbb{R}^{d_h} : d_{\mathbb{R}^{d_h}}(\boldsymbol{h}, \widetilde{\mathcal{H}}) \le \epsilon_{\mathcal{H}}\},$$

*where $0 < \epsilon_{\mathcal{H}} < \min\{\tau_{\widetilde{\mathcal{H}}}, 1\}$. Equivalently for $\mathcal{X}$,*

$$\mathcal{X} = \{\boldsymbol{x} \in \mathbb{R}^{d_x} : d_{\mathbb{R}^{d_x}}(\boldsymbol{x}, \widetilde{\mathcal{X}}) \le \epsilon_{\mathcal{X}}\},$$

*where $0 < \epsilon_{\mathcal{X}} < \min\{\tau_{\widetilde{\mathcal{X}}}, 1\}$.*

*Remark* 2. This restriction to the type of state and input spaces we consider is sufficient to construct all neural networks of interest by choosing pair of points from the space in question, and construct the weight and bias as in Equation (4). It is also argued in Bolager et al. (2023) that most interesting real-world application will contain some noise and make sure that the state and input spaces is approximately on the form given in the definition.

As we are not considering the eigenfunctions of the system, only prediction, and we are working with real valued neural networks, we are in the setting with $\mathbb{F} = \mathbb{R}$ in Equation (16).

## B.1 Uncontrolled systems

For uncontrolled systems the evolution can be described by $\boldsymbol{h}_t = F(\boldsymbol{h}_{t-1})$. For the theory developed in this section, we require the following assumptions.

**Assumption 3.** We assume the measure $\mu_h$ on the space $\mathcal{F} = L^2(\mathcal{H}, \mu_h)$ is regular and finite for compact subsets.

**Assumption 4.** The following assumptions is made for the dictionary and the Koopman operator $\mathcal{K}$ associated to the dynamical system defined by the map $F$:

4.1. Any dictionary $\Psi_M$ satisfies $\mu_h\{\boldsymbol{h} \in \mathcal{H} \mid \boldsymbol{c}^\mathsf{T}\Psi_M(\boldsymbol{h}) = 0\} = 0$, for all nonzero $\boldsymbol{c} \in \mathbb{R}^M$.

4.2. $\mathcal{K}: \mathcal{F} \to \mathcal{F}$ is a bounded operator.

Assumption 3 is not very limited, as it holds for most measures we are interested in, such as measures absolutely continuous to the Lebesgue measure. For Assumption 4.1 we have the following result.

**Lemma 1.** *Let $(\mathbb{R}^{d_h}, \mathscr{B}(\mathbb{R}^{d_h}), \mu_h)$ be a measurable space with $supp(\mu_h) = \mathcal{H}$, and $\lambda$ be the Lebesgue measure for $\mathbb{R}^{d_h}$. If for all non-zero $\boldsymbol{c} \in \mathbb{R}^{d_h}$, the following holds:*

- *The set of functions $\{\psi_1, \dots, \psi_M\}$ is linearly independent.*

- *$\boldsymbol{c}^\mathsf{T}\Psi_M = \sum_{m=1}^M c_m\psi_m$ is analytic on $\mathbb{R}^{d_h}$,*

- *$\mu_h \ll \lambda$,*

*then Assumption 4.1 is satisfied. In particular, it is satisfied when $\{\psi_i\}_{m=1}^M$ are independent tanh functions.*

*Proof.* Let $\boldsymbol{c} \in \mathbb{R}^{d_h}$ be any non-zero vector. As $\{\psi_1, \dots, \psi_M\}$ are linear independent, we have $\boldsymbol{c}^\mathsf{T}\Psi_M \not\equiv 0$. As $\boldsymbol{c}^\mathsf{T}\Psi_M$ is analytic, the set

$$A = \{\boldsymbol{h} \in \mathbb{R}^{d_h} \mid \boldsymbol{c}^\mathsf{T}\Psi_M(\boldsymbol{h}) = 0\}$$

has measure zero, so its Lebesgue measure $\lambda(A) = 0$ due to Proposition 1 in Mityagin (2020). We also have $\lambda(A \cap \mathcal{H}) \leq \lambda(A) = 0$. Finally due to absolute continuity of the measure $\mu_h$ w.r.t. $\lambda$, we have

$$\mu_h(\{\boldsymbol{h} \in \mathcal{H} \colon \boldsymbol{c}^\mathsf{T} \Psi_M(\boldsymbol{h}) = 0\}) = \lambda(A \cap \mathcal{H}) = 0,$$

and hence Assumption 4.1 holds. As the linear projection and shift of bias is analytic on $\mathbb{R}^{d_h}$, tanh is analytic on $\mathbb{R}$, and analytic functions are closed under compositions, means Assumption 4.1 holds when $\Psi_M$ is a set of linearly independent neurons with tanh activation function. $\qquad\square$

*Remark* 3. The requirement of the functions being analytic for the whole $\mathbb{R}^{d_h}$ can certainly be relaxed if necessary to an open and connected set $U$, s.t. $\mathcal{H} \subseteq U$). Further relaxation can be made with some additional work. The result above also agrees with the claim made in Korda & Mezić (2018b) about Assumption 4.1 holds for many measures and most basis functions such as polynomials and radial basis functions. Finally, we note that the independence requirement is easily true when we sample the neurons.

Assumption 4.2 is commonly enforced in Koopman theory when considering convergence of EDMD and for example holds when $F$ is Lipschitz, has Lipschitz invertible, and $\mu_h$ is the Lebesgue measure (Korda & Mezić, 2018b).

We note

$$\mathcal{NN}_{[1,1:\infty]} = \bigcup_{M=1}^{\infty} \mathcal{NN}_1(\mathcal{H}, \mathbb{R}^M)$$

is the space of all hidden layers with tanh activation function from $\mathcal{H}$. We continue with a result relating this space with $\mathcal{F}$.

**Lemma 2.** *For a tanh activation function and when Assumption 3 holds, then $\mathcal{NN}_{[1,1:\infty]}$ is dense in $\mathcal{F}$ and has a countable basis $\{\psi_i \in \mathcal{NN}_{[1,1:\infty]}\}_{i=1}^{\infty}$, both when the parameter space is the full Euclidean space and when constructed as in Equation* (4).

*Proof.* It is well known that such a space is dense in $C(\mathcal{H}, \mathbb{R}^{d_y})$ for any $d_y \in \mathbb{N}_{>0}$. This holds both when the weight space is the full Euclidean space (Cybenko, 1989; Pinkus, 1999) and when limited to the weight construction in Equation (4) (Bolager et al., 2023). As $\mathcal{H}$ is compact, we have that $\mathcal{NN}_{[1,1:\infty]}$ is dense in $\mathcal{F}$. Furthermore, as $\mathcal{F}$ is a separable Hilbert space and metric space, there exists a countable subset $\{\psi_i \in \mathcal{NN}_{[1,1:\infty]}\}_{i=1}^{\infty}$ that is a basis for $\mathcal{F}$. $\qquad\square$

The following lemma makes sure we can circumvent assumptions made in Korda & Mezić (2018b), which requires on the dictionary in the EDMD algorithm to be an orthonormal basis (o.n.b.) of $\mathcal{F}$.

**Lemma 3.** *Let $H, H'$ be the dataset used in Equation* (19). *For every set of $M \in \mathbb{N}$ linearly independent functions $\Psi_M = \{\phi_i\}_{i=1}^{M}$ from a dense subset of $\mathcal{F}$ and any function $f = c^\mathsf{T}\Psi_M$, there exists a $\tilde{c}$ such that*

$$\tilde{c}^\mathsf{T} \tilde{K} \tilde{\Psi}_M = c^\mathsf{T} K \Psi_M$$

*and*

$$\tilde{c}^\mathsf{T} \tilde{\Psi}_M = f = c^\mathsf{T} \Psi_M,$$

*where $\tilde{\Psi}_M = [\tilde{\psi}_1, \tilde{\psi}_2, \ldots, \tilde{\psi}_M]$ are functions from an orthornormal basis $\{\tilde{\psi}_i\}_{i=1}^{\infty}$ of $\mathcal{F}$, and $K, \tilde{K}$ are the Koopman approximations for the dictionaries $\Psi_M$ and $\tilde{\Psi}_M$ respectively.*

*Proof.* As $\mathcal{F}$ is a separable Hilbert space and a metric space, there exists a countable basis $\{\phi_i\}_{i=1}^{M} \cup \{\phi_i\}_{i=M+1}^{\infty}$, and by applying the Gram-Schmidt process to the basis, we have an o.n.b. $\{\tilde{\psi}_i\}_{i=1}^{\infty}$. Any $M$ step Gram-Schmidt process applied to a finite set of linearly independent vectors, can be written as a sequence of invertible matrices $V = \prod_{j=1}^{M+1} V_j$. Each matrix $V_j$ for $j < M+1$ transforms the $j$th vector and

the last matrix simply scales. Constructing such matrix $V$ applied to $\Psi_M$ yields $\tilde{\Psi}_M$. Setting $\tilde{c}^\mathsf{T} = c^\mathsf{T} V^{-1}$, which means

$$\tilde{c}^\mathsf{T} \tilde{\Psi}_M = c^\mathsf{T} V^{-1} \tilde{\Psi}_M = c^\mathsf{T} \Psi_M = f.$$

Furthermore, we have

$$\begin{aligned}
\tilde{c}^\mathsf{T} \tilde{K} \tilde{\Psi}_M &= c^\mathsf{T} V^{-1} [\tilde{\Psi}_M(H') \tilde{\Psi}_M(H)^+] V \Psi_M \\
&= c^\mathsf{T} V^{-1} [V \Psi_M(H') \Psi_M(H)^+ V^{-1}] V \Psi_M \\
&= c^\mathsf{T} [\Psi_M(H') \Psi_M(H)^+] \Psi_M = c^\mathsf{T} K \Psi_M.
\end{aligned}$$

$\square$

We are now ready to prove Theorem 2 from the paper, namely the existence of networks for finite horizon predictions. We denote $\mathcal{F}^{d_y}$ as the space of vector valued functions functions $f = [f_1, f_2, \ldots, f_{d_y}]$, where $f_i \in \mathcal{F}$ and $\|f\| = \sum_{i=1}^{d_y} \|f_i\|_{L^2}$. We also recall that $\mathcal{F}_M$ is one hidden layer with *tanh* activation function, which is equivalent to saying the dictionary are $M$ neurons. In addition, we let $K_N$ be the Koopman approximation for $\mathcal{F}_M$ where $N$ data points have been used to solve the least square problem.

**Theorem 5.** *Let $f \in \mathcal{F}^{d_y}$, $H, H'$ be the dataset with $N$ data points used in Equation* (19)*, and Assumption 3 and Assumption 4 hold. For any $\epsilon > 0$ and $T \in \mathbb{N}$, there exist an $M \in \mathbb{N}$ and hidden layers $\mathcal{F}_M$ with $M$ neurons and matrices $C$ such that*

$$\lim_{N \to \infty} \int_{\mathcal{H}} \|C K_N^t \mathcal{F}_M - f \circ F^t\|_2^2 d\mu_h < \epsilon, \tag{21}$$

*for all $t \in [1, 2, \ldots, T]$. In particular, there exist hidden layers and matrices $C$ such that*

$$\lim_{N \to \infty} \int_{\mathcal{H}} \|C K_N^t \mathcal{F}_M - F^t\|_2^2 d\mu_h < \epsilon. \tag{22}$$

*Proof.* W.l.o.g., we let $d_y = 1$ and uses vector $c$ instead of matrix $C$ in the proof. Due to Lemma 2, we know there exist hidden layers $\mathcal{F}_M$ and vectors $c$ such that $\|f_m - f\|_{L^2}^2 < \epsilon_2$, where $c^\mathsf{T} \mathcal{F}_M = f_m$ and

$$\epsilon_2 < \frac{\epsilon}{2 \cdot \max\{\|\mathcal{K}\|_{op}^{2T}, \|\mathcal{K}\|_{op}^2\}},$$

with $\|\cdot\|_{op}$ being the operator norm. This is possible due to Assumption 4 and Definition 2. We then have for any $t \in [1, 2, \ldots, T]$

$$\begin{aligned}
&\lim_{N \to \infty} \int_{\mathcal{H}} \|c^\mathsf{T} K_N^t \mathcal{F}_M - \mathcal{K}^t f\|_2^2 d\mu_h \\
&\leq \lim_{N \to \infty} \int_{\mathcal{H}} \|c^\mathsf{T} K_N^t \mathcal{F}_M - \mathcal{K}^t f_m\|_2^2 d\mu_h + \|\mathcal{K}^t f_m - \mathcal{K}^t f\|_{L^2}^2 \\
&\leq \lim_{N \to \infty} \int_{\mathcal{H}} \|c^\mathsf{T} K_N^t \mathcal{F}_M - \mathcal{K}^t f_m\|_2^2 d\mu_h + \|f_m - f\|_{L^2}^2 \max\{\|\mathcal{K}\|_{op}^{2T}, \|\mathcal{K}\|_{op}^2\} \\
&< \frac{\epsilon}{2} + \frac{\epsilon}{2} = \epsilon,
\end{aligned}$$

where we use Theorem 5 in Korda & Mezić (2018b) to bound $\|c^\mathsf{T} K_N^t \mathcal{F}_M - \mathcal{K}^t f_m\|_2^2 d\mu_h$; we might need a larger $M$, which we simply set and the bound of $f_m - f$ still holds. From the convergence above, Equation (21) follows by definition of the Koopman operator. For Equation (22), we note that $f(\boldsymbol{h}) = \boldsymbol{h}$ is in $\mathcal{F}^{d_h}$ due to Definition 2, and the result holds. $\square$

### B.2 Controlled systems

The results above cannot easily be shown for controlled systems. The reason is that the dictionary space one uses is not a basis for the observables in the controlled setting, with the dynamical system extended by the left-shift operator, and the simplification made for EDMD in controlled systems. The results above may be extended, but EDMD will not converge to the Koopman operator, but rather to $P_\infty^\mu \mathcal{K}_{\mathcal{F}_\infty}$, where $P_\infty^\mu$ is the $L^2(\mu)$ projection onto the closure of the dictionary space (Korda & Mezić, 2018a). However, results exist for continuous controlled systems, with certain convergence results for the generator (Nüske et al., 2023). This is not a result that is as strong as in the uncontrolled setting, but an interesting path to connect RNNs/NeuralODEs to such theory.

## C   Evaluation measures

### C.1   Geometrical measure

The Kullback-Leibler divergence of two probability densities $p(\boldsymbol{x})$ and $q(\boldsymbol{x})$ is defined as

$$D_{KL}(p(\boldsymbol{x})\|q(\boldsymbol{x})) = \int_{\boldsymbol{x}\in\mathbb{R}} p(\boldsymbol{x}) \log \frac{p(\boldsymbol{x})}{q(\boldsymbol{x})} d\boldsymbol{x}. \tag{23}$$

In order to be able to accurately evaluate also high-dimensional systems, we follow the approach used in Hess et al. (2023) and place Gaussian Mixture Models (GMM) on the along the true trajectory $\boldsymbol{x}$ and predicted $\hat{\boldsymbol{x}}$ trajectories, obtaining $\hat{p}(\boldsymbol{x}) = \frac{1}{T}\sum_{t=1}^{T}\mathcal{N}(\boldsymbol{x},\boldsymbol{x}_t,\Sigma)$ and $\hat{q}(\boldsymbol{x}) = \frac{1}{T}\sum_{t=1}^{T}\mathcal{N}(\boldsymbol{x},\hat{\boldsymbol{x}}_t,\Sigma)$ for $T$ snapshots. Using the estimated densities, we consider a Monte Carlo approximation of Equation (23) by drawing $n$ random samples from the GMMs and obtain the density measure

$$D_{KL}(\hat{p}(\boldsymbol{x})\|\hat{q}(\boldsymbol{x})) \approx \frac{1}{n}\sum_{i=1}^{n} \log \frac{\hat{p}(\boldsymbol{x})}{\hat{q}(\boldsymbol{x})}. \tag{24}$$

We call this metric empirical KL divergence (EKL) in the manuscript. To make our results comparable with Hess et al. (2023), we use $\sigma^2 = 1.0$ and $n = 1000$.

## D   Model details and comparison

### D.1   KIRNN

The implementation of our Koopman-informed RNNs was done in Python since the key tools for the algorithm already exist as Python libraries. The algorithm requires the ability to sample weights and biases, thus we used the Python library `swimnetworks` by Bolager et al. (2023). Furthermore, we were able to alleviate the approximation of the Koopman operator, in the uncontrolled as well as controlled setting using some functionalities from the Python library `datafold` by Lehmberg et al. (2020).

KIRNNs have only a few hyperparameters: the number of nodes in the hidden layer, the activation function of the hidden layer, and a cutoff for small singular values in the least-squares solver. We refer to the singular value cutoff hyperparameter as the regularization rate. In the case of a KIRNN with time delay, additional hyperparameters are the number of time delays and the number of PCA components, if used. KIRNNs do not only have a low fit time but also a short hyperparameter tuning since there are only a few degrees of freedom. Thus, our newly proposed method offers efficiency beyond the short training time.

The Van der Pol experiment from Section 3.2 is one of the simplest demonstrations of the KIRNN. Our initial experiments used a smaller training dataset than the one described in Section 3.2, with very short trajectories consisting of only three time steps. Wit KIRNN, we were able to approximate the dynamics from this data very well. However, it was not possible to train the gradient-based shPLRNN with such a dataset; thus, we used longer trajectories to be able to compare our method with it. The final choice of hyperparameters for the hyperparameters is given in Table 3.

In addition to the experiments already discussed in the main text, we were interested to see if our proposed method has a tendency to overfit on the training data. Thus, we investigated the effect of increasing the number of neurons and evaluated our model on training and validation data. Results from five runs are shown in Figure 11. We notice that the errors decrease up to a certain number of neurons and stagnate afterwards, and as the gap between the training and validation error only grows slightly with the number of parameters we believe that overfitting does not occur.

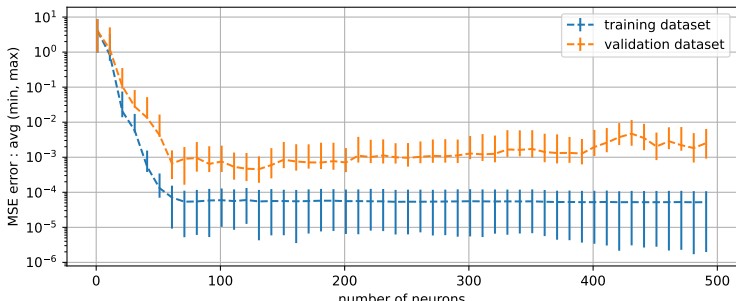

Figure 11: Training and validation MSE error over number of neurons for the Van der Pol experiment from Section 3.2.

Training the 1D Van der Pol from Section 3.3 was done similarly, the number of hidden nodes and activation function remained unchanged. The hyperparameters are listed in Table 3. A time delay of six was chosen, however it was also possible to use only two time delays and get an excellent performance, but we opted against this choice because this was based on our knowledge that the true system is two dimensional. Thus, we opted for a higher number of time delays which then get reduced with the additional PCA transformation to mimic a real-world scenario where the true dimension of the problem is unknown.

The chaotic systems from Section 3.4 required more hidden nodes in order to obtain good prediction, as compared to the Van der Pol but nonetheless the process of hyperparameter tuning was simple. The final choice of hyperparameters for the Lorenz and Rössler problems is given in Table 3.

With the forced Van der Pol example, presented in Section 3.6, we use the KIRNN as a surrogate linear model with an LQR controller. In order to be able to accurately control the state, the surrogate needs to capture the dynamics of the system with sufficient accuracy. The hyperparameters are listed in Table 3. We opted for an LQR controller as it is efficient and a good choice for linear systems. For our LQR, the costs are set as $R = 1$, and $Q = \mathrm{diag}(10)$. The dataset we use contains randomly initialized trajectories evolved in time, as well as randomly chosen control inputs. We observed that a dataset with many shorter trajectories is more useful than a dataset with a few longer trajectories since for this problem diverse initial conditions are more informative of the vector field of the system, as opposed to long trajectories which become periodic. The necessary effort for tuning the KIRNN hyperparameters and LQR controller parameters was low.

The training of a KIRNN on weather data was performed using time-delay embeddings to account for a possibly partial observation of the state. The objective is to train a model which can predict the temperature. To find the hyperparameters a grid search was performed, and this is documented in Table 4. The final choice of hyperparameters is given in Table 3. After training, predictions are done for fixed chunks of time, which are concatenated together afterwards. The number of time-delays dictates how many ground-truth datapoints are necessary to predict the next state, which gets concatenated with the previous predictions. We see this as a reasonable approach for weather prediction, as typically one would use the available information for a fairly long period of time, and predict for a fixed, likely shorter time horizon.

For the electricity consumption data the training setup was the same as the one with weather data, thus we do not elaborate further on this. In Table 4 and Table 5 we detail the hyperparameter grid search for Section 3.7 performed for the KIRNN and ESN models.

### D.1.1 Hyperparameters

See Table 3 for an overview of the final hyperparameters for all KIRNN models.

Table 3: Hyperparameters of KIRNN models

|  | Van der Pol 3.2 | 1D Van der Pol 3.2 | Lorenz 3.4 | Rössler 3.4 | forced Van der Pol 3.6 | Weather 3.7 | Electricity consumption 3.7 |
|---|---|---|---|---|---|---|---|
| Hidden layer width | 80 | 80 | 200 | 300 | 128 | 256 | 64 |
| Activation function | *tanh* | *tanh* | *tanh* | *tanh* | *tanh* | *tanh* | *tanh* |
| Regularization rate | 1e-8 | 0 | 1e-7 | 1e-4 | 1e-10 | 1e-6 | 1e-8 |
| Time delays | - | 6 | - | - | - | 168 | 240 |
| PCA components | - | 2 | - | - | - | - | - |

Table 4: Hyperparameters used in the grid search for training KIRNN and ESN for the weather prediction in Section 3.7.

|  | KIRNN | ESN |
|---|---|---|
| Width/units | $32, 64, 128$ | $16, 32, 64, 128, 256, 512$ |
| Regularization rate | 1e-10, 1e-8, 1e-6, 1e-4, 1e-2 | — |
| Leak rate | — | $0.01, 0.1, 0.3, 0.5, 0.7, 1.0$ |
| Spectral radius | — | $0.1, 0.5, 1.0$ |

Table 5: Hyperparameters used in the grid search for training KIRNN and ESN for the electricity consumption prediction in Section 3.7.

|  | KIRNN | ESN |
|---|---|---|
| Width/units | $32, 64, 128, 256, 512$ | $32, 64, 128, 256, 512$ |
| Regularization rate | 1e-8, 1e-6, 1e-4, 1e-2 | — |
| Leak rate | — | $0.01, 0.1, 0.3, 0.5, 0.7, 1.0$ |
| Spectral radius | — | $0.1$ |

### D.1.2 Hardware

The machine used for training the KIRNNs was 13th Gen Intel(R) Core(TM) i5-1335U @ 4.6 GHz (16GB RAM, 12 cores), no GPU hardware was used.

For the experiments with the real-world data in Section 3.7, we used a machine with AMD EPYC 7402 @ 2.80GHz (256GB RAM, 24 cores).

### D.2 ESN

For the implementation of ESNs we used the Python library `reservoirpy` by Trouvain et al. (2020). All models were trained using a single reservoir and a ridge regression readout. For the synthetic datasets, we consider only reservoir models without any warm-up phase in order to keep them comparable to our KIRNN which also does not have a warm-up.

For the chaotic systems, we were able to find guidelines in the literature for a suitable choice of hyperparameters specifically tailored to the Lorenz system (see (Viehweg et al., 2023)). With minor modifications and following the proposed guidelines, we found hyperparameters also for the Rössler system. The choice of hyperparameters is specified in Table 6, any hyperparameters not mentioned are set to the default value of `reservoirpy`.

For the Van der Pol problem many hyperparameter combinations were tried out until we were able to find a reservoir computing model with good performance and sufficient robustness to a change in the random seed. The hyperparameter combinations we considered are shown in Table 7. Due to the high expenses of a grid-search approach, a random search was employed and 1000 hyperparameter combinations were considered. The hyperparameter choice with the best performance on the validation dataset was selected and then evaluated on the test dataset. The final choice of hyperparameters is documented in Table 6, and any unmentioned hyperparameters are assumed to be set to the default value from the `reservoirpy` library.

### D.2.1 Hyperparameters

Table 6: Hyperparameters of reservoir models

|  | Van der Pol 3.2 | Lorenz 3.4 | Rössler 3.4 | Weather 3.7 | Electricity consumption 3.7 |
|---|---|---|---|---|---|
| Width/units | 500 | 300 | 500 | 32 | 32 |
| Leak rate | 0.9 | 0.3 | 0.3 | 0.01 | 0.3 |
| Spectral radius | 0.5 | 1.25 | 0.5 | 0.1 | 0.1 |
| Input scaling | 0.05 | 0.1 | 0.1 | 1 | 1 |
| Connectivity | 0.8 | 0.1 | 0.1 | 0.5 | 0.5 |
| Inter connectivity | 0.2 | 0.2 | 0.2 |  |  |
| Ridge regularization coeff. | 1e-10 | 1e-4 | 1e-8 | 1e-6 | 1e-6 |
| Warmup steps | 0 | 0 | 0 | 168 | 240 |

Table 7: Hyperparameters used in random search on a grid for a Van der Pol reservoir model

|  | Van der Pol 3.2 |
|---|---|
| Width/units | $100, 200, 500$ |
| Leak rate | $0.1, 0.3, 0.5, 0.7, 0.9$ |
| Spectral radius | $0.25, 0.5, 0.75, 1, 1.25, 1.5, 2, 3, 5$ |
| Input scaling | $0.05, 0.1, 0.5, 1, 1.5, 2$ |
| Connectivity | $0.2, 0.4, 0.6, 0.8, 1$ |
| Inter connectivity | $0.2, 0.4, 0.6, 0.8, 1$ |
| Ridge regularization coeff. | 1e-4, 1e-6, 1e-8, 1e-10 |
| Warmup steps | 0 |

### D.2.2 Hardware

The machine used for fitting the ESN models was 13th Gen Intel(R) Core(TM) i5-1335U @ 4.6 GHz (16GB RAM, 12 cores).

### D.3 shPLRNN

For an explanation of shPLRNN, see Section 3.1.1.

### D.3.1 Hyperparameters

We used the clipped shPLRNN trained by GTF. For the Van der Pol, Lorenz and the weather datasets we considered a fixed GTF parameter $\alpha$, while for the Rössler system we considered an adaptive $\alpha$ (starting

from an upper bound) as proposed by (Hess et al., 2023). The code repository by (Hess et al., 2023) was used to perform the computations. The hyperparameters selected for all datasets are detailed in Table 8. Any hyperparameters not specified are set to their default values in the corresponding repository of (Hess et al., 2023). For the Rössler dataset, finding optimal hyperparameters was more challenging, and for training, we also utilized regularizations for the latent and observation models.

Table 8: Hyperparameters of shPLRNN trained by GTF

|  | Van der Pol 3.2 | Lorenz 3.4 | Rössler3.4 | Weather 3.7 | Electricity consumption 3.7 |
|---|---|---|---|---|---|
| Hidden dimension | 35 | 100 | 50 | 300 | 900 |
| Number of hidden layers | 3 | 3 | 3 | 3 | 4 |
| Batch Size | 32 | 30 | 50 | 32 | 32 |
| Sequence length | 37 | 100 | 150 | 100 | 100 |
| Epochs | 1300 | 2000 | 2000 | 1000 | 1500 |
| GTF parameter ($\alpha$) | 0.98 | 0.3 | 0.9 (Upper bound) | 0.94 | 1 (Upper bound) |
| Latent model regularization rate | - | - | 1e-6 | - | 1e-4 |
| Observation model regularization rate | - | - | 1e-4 | - | 1e-4 |

### D.3.2   Hardware

The hardware we used to iteratively train the clipped shPLRNN models includes an 11th Gen Intel(R) Core(TM) i7-11800H CPU @ 2.30GHz and 64.0 GB of RAM (63.7 GB usable).

