# OpenReview forum: "Koopman-informed recurrent neural networks"
_TMLR — Accepted by TMLR_

### Review · Reviewer_NYv7 · 2026-04-24

**Summary Of Contributions:**

### Overall assessment

A model for time-series data based on the linear dynamics described by the Koopman operator is proposed. The core idea is to use neural nets with randomly (yet adaptively to data) sampled weights as observables for the Koopman operator.

The paper is basically easy to follow, the proposed method is technically reasonable, and the empirical results are convincing, at least for showcasing that the proposed method somewhat works on various dynamics.

### Specific comments

(1) I am not sure how it is justifiable to put the proposed method in the context of RNNs. The proposed method requires the hidden states, $h$, to be fully observed, which I think is not the case in many use cases of RNNs generally. In my understanding, the proposed method is effectively an instance of EDMD (with control) with the randomly sampled neural nets used as observables. More clarification on how we should think of the fully observed "hidden" states would be highly helpful.

(2) Related to (1), what are the "hidden" states of KIRNN in the experiments? Are they simply the original time-series data? It should be stated somewhere.

(3) The motivation for discussing the control in Section 2.3 is unclear. LQR with (E)DMD is indeed possible and has been actively studied / used by many researchers. But merely being possible does not usually become a good reason to do so.

(4) Although shPLRNN is chosen as a baseline of RNN, the datasets used in the original paper of shPLRNN are not examined now. Do you have any results on them? Such a direct comparison would be, maybe not necessary, but certainly useful for readers.

(5) In Section 3.5: "Instead of DMD/EDMD we employ the KIRNN to model the dynamics ..." What is the difference between EDMD assumed here and KIRNN? As I commented in (1), KIRNN looks like EDMD itself.

Minor points:
- Fourth line of Section 2.1: "$F_\mathcal{H}$ and $F_\mathcal{X}$" ... I couldn't find the exact definition of these.
- Line 6 of Algorithm 2: Why $\mathcal{F}_M(H')$ is here?

**Audience:**

Yes

**Audience Explanation:**

Time-series / dynamics prediction is one of the most ubiquitous tasks in science and engineering. Moreover the proposed method is simple, convenient in training, which makes it attractive in many applications.

**Claims And Evidence:**

Yes

**Claims Explanation:**

The empirical results showcase various dynamics and are somewhat convincing in showing that the proposed method indeed works on them. The experiments are not very targeted to specific questions, but I think it is fine, as the paper's claim lies in showing the possibility of the RNN-like model without gradient descent.

**Requested Changes:**

On comment (1) above, the paper would need more clarification on how it is justifiable to think "hidden" states $h$ of an RNN are fully observed.

---

> ### Author Response · Authors · 2026-05-08
> **Response to Reviewer NYv7**
>
> We would like to thank the reviewer for the detailed feedback and appreciate their strong understanding of the EDMD approach. Below we try to address all comments individually, and we are open to any follow-up or further feedback.
>
> **Answers to specific comments:**
>
> (1) Our construction of KIRNN satisfies the general definition of an RNN, which is based on recurrence. This allows a single neural network to be used iteratively, instead of a separate one for each time step. In particular, we study RNNs applied to learn dynamical systems as in \[1\] and \[2\]. We use the Koopman operator/EDMD connection to RNNs to address the core challenge of our research: finding ways to avoid problems associated with backpropagation for RNNs, especially for chaotic systems. We argue that this connection makes the paper valuable to both the EDMD and RNN communities.
>
> Concerning the hidden states: In RNNs, this hidden state $h_0$ is typically initialized to a default value, such as the zero vector, and then one uses the first $t_i$ points in the input sequence to create a state $h_i$. Starting with this state $h_i$, one then can produce observables and predictions for $t>t_i$. This is often referred to as the “warm-up phase” \[3\] or the “encoder part” of the RNN \[4\]. To make sure the $h_i$ captures the relevant information such that one can produce accurate outputs for $t>t_i$, one usually propagates errors / gradients backwards through the warm-up phase. However, this is not the only way to set up hidden states. To see how one can apply KIRNN in these settings, we first note that the input need not be the full state $h$, but rather any observable of the state (e.g. the observable values during the warmup phase). Whether this is one coordinate of the state, the mean of its dimensions, or a nonlinear function of the state depends on the context. The important part is that the input contains some information about the state of the dynamical system, and how large $t_i$ is, that is, how long the warm-up phase must be, depends on how much information about the full state each input point contains. This is exactly what we did, for example, in the experiment in subsection 3.3, where we have partial observations, and $t_i$ is then the time delay used. So as long as one has $t_i$ observables from the dynamical system, similar to what is needed for RNNs trained by gradient-based methods, one can use these observables to approximate the Koopman part of KIRNN, and therefore apply KIRNN also in the setting where we do not have access to the full state.
>
> To conclude, the KIRNN architecture is indeed an RNN, and it can also be applied to settings where we do not have access to the full state $h$ during training, as demonstrated in the partial observation experiment. We focus more on the full-state setting to mitigate the issues for chaotic systems as discussed in \[2\], but the approach is certainly not limited to this.
>
> (2) Yes, in most experiments the data contains a state and the state after one time step. For experiments in Sections 3.3 and 3.7 this is not the case. In the first case we have partial observations (only the first coordinate is contained), while in the latter case we have real-world observations of unknown systems thus we cannot know if the full state is contained. We have updated the first paragraph of Section 3 of the manuscript with a clarification.
>
> (3) The addition of control was motivated by the fact that we focus on the connection to RNNs, which cannot handle nonlinear control problems through linear control theory. For readers with a background in EDMD, the inclusion of optimal control can be seen as an attempt to empirically investigate whether the sampling of the dictionary is useful. We also include additional details on LQR to provide the necessary background in optimal control theory for unfamiliar readers and to show that we can rewrite the resulting model as an RNN.
>
> (4) Indeed, only the Lorenz-63 experiment from the shPLRNN paper is done in our experiments. We did not include the experiments on real-world data from the shPLRNN paper because the data from ECG and EEG measurements are intrinsically high-dimensional. As we mentioned in the conclusion, one limitation of our approach is its poor performance on high-dimensional problems.
>
> (5) We agree that the phrasing is confusing and could benefit from clarification. Since Kern et al. use well-known dictionaries for EDMD in their experiments, we only intended to differentiate between the dictionary choices and to emphasize that we use sampled neurons in our experiments. We have updated the manuscript accordingly.
>
> **Answers to minor points:**
>
> - Yes, these definitions appear to be missing. We have added the following to the paper: $ F\_{\\mathcal{H}} = C_h \\mathcal{F}\_M $ and similarly $ F\_{\\mathcal{X}} = C_x \\mathcal{G}\_{\\hat{M}} $.
>
> - There is a typo in the algorithm. Thank you for pointing this out. Now we have removed $ \\mathcal{F}\_M (H’) $.

---

> > ### Author Response · Authors · 2026-05-08
> > **References**
> >
> > \[1\] Gajamannage, K., Jayathilake, D. I., Park, Y., & Bollt, E. M. “Recurrent neural networks for dynamical systems: Applications to ordinary differential equations, collective motion, and hydrological modeling”. Chaos: An Interdisciplinary Journal of Nonlinear Science, 33(1). 2023.
> >
> > \[2\] Hess, F., Monfared, Z., Brenner, M., & Durstewitz, D. “Generalized Teacher Forcing for Learning Chaotic Dynamics”. In International Conference on Machine Learning. 2023
> >
> > \[3\] TensorFlow. “Tutorial on time series forecasting”. URL https://www.tensorflow.org/tutorials/structured_data/time_series. Accessed on 07.05.2026.
> >
> > \[4\] Bishop, C. M., & Bishop, H. “Deep learning: Foundations and concepts”. Springer Nature. 2023

---

> > ### Comment · Reviewer_NYv7 · 2026-05-21
> >
> > Thank you for the response.
> >
> > As for points (1), (2), (5), my concern is about how you present the method. The proposed training procedures in Algorithm 2 require not only time-series inputs $x$ and outputs $y$ but also the states $h$. On the other hand, ordinary RNN training requires only $x$ and $y$, because $h$ is computed within the model as *hidden* states. Such a fundamental difference is explained only implicitly.
> > If $h$ is set as the time-series data or partial observation from them, then what is $y$? Is it correct to think that when $h$ is set as "the full state", you have $h=y$?  Such clarification would be necessary.

---

> > > ### Author Response · Authors · 2026-05-24
> > >
> > > Thank you for the additional comment, we believe that it helped us better understand what was meant in the review. We agree that it is confusing to use both h and x as model inputs, and it is not typical for RNNs. We have updated the manuscript with an additional paragraph before the beginning of section 2.1 (page 3 and start of page 4, changes are as always in blue).
> > >
> > > We further want to clarify our perspective on hidden states:
> > > In most cases the hidden states of RNNs are determined via the process of iterative optimization and are dynamically changing during model training. In our case, we do not perform iterative optimization, thus we also do not iterate over different hidden state representations. Instead, we explicitly use either the full state (if available) or time delays of partial observation (as surrogate for the state) and convert it to higher-dimensional representations with sampled neurons (corresponding to dictionary evaluation in the EDMD framework). Indeed, as described now also in the manuscript, if the full state is available and should also be observed , then h=y as mentioned by the reviewer.

---

> > > > ### Comment · Reviewer_NYv7 · 2026-05-29
> > > >
> > > > Thanks for the comment and the revision! The added explanation makes sense. Although I'm still not fully convinced about the motivation to put the method in the RNN context, anyway the paper looks much less confusing now.

---

### Review · Reviewer_RM6F · 2026-04-26

**Summary Of Contributions:**

This paper proposes a Koopman-informed recurrent neural network for learning dynamical systems from data. The main idea is to avoid backpropagation-through-time by first constructing a random nonlinear feature map and then fitting the remaining linear components with least-squares/EDMD-type computations.

The resulting model can be viewed as a recurrent architecture with three parts: a nonlinear lifting map, a linear evolution operator in the lifted space, and a projection back to the state or observation space. This structure gives the model a connection to Koopman operator theory and allows the authors to use tools such as spectral analysis and, in the controlled setting, LQR-style control.

**Audience:**

Yes

**Audience Explanation:**

Yes. The paper should be of interest to part of the TMLR audience, especially readers working on dynamical systems, Koopman operator methods, recurrent models, and efficient training alternatives to BPTT. The combination of random features, EDMD, and RNN-style rollout is relevant.

**Broader Impact Concerns:**

N.A

**Claims And Evidence:**

Yes

**Claims Explanation:**

The experiments provide reasonable evidence that the method can train faster than gradient-based RNN baselines while achieving competitive performance on the tested dynamical systems. A proof is provided for the theorem.

**Requested Changes:**

The following are my questions and comments:

1. The data-dependent sampling rule is intuitive. Could the authors clarify whether there is any probabilistic guarantee for the proposed sampling strategy, or whether it is mainly a heuristic supported by experiments?

2. Did the authors use all the data in approximating $K$, even in the real-world data experiment? Computing the pseudoinverse does not scale very well when N and M are large.

3. Theorem 2 analyzes the approximation error of $CK_N^tF_M$. However, the actual procedure repeatedly does projection and re-lifting. Can the authors provide a theoretical analysis for this procedure?

4. Section 2.1 introduces the direct sampled-RNN regression in Eq. (6), while Section 2.2 imposes the Koopman factorization. Although it allows for linear control, can the authors comment whether the Koopman structure improves prediction accuracy over the simple least squares in (6)?

Minor comment:

1. The main text uses $C = HF_M(H)^{+}$.  Please check the expression for $C$ in Algorithm 2.

---

> ### Author Response · Authors · 2026-05-08
> **Response to Reviewer RM6F**
>
> We are grateful for the feedback provided in this review, we appreciate the reviewer for spending time and effort to understand our work. We try to directly address the concerns with this response, and have made changes to the manuscript based on this feedback. If there are follow-up questions or further concerns we would be happy to address them.
>
> **Answers to questions and comments:**
>
> 1. Theoretical results concerning the sampling method used in the paper can be found in \[1\] and \[2\]. In \[1\], the original paper in which the authors describe the sampling strategy, they also present universal approximation results and convergence results for certain function spaces. These are, however, not probabilistic. For probabilistic bounds, one can consider general probabilistic results on sampling neural networks, such as \[2\], and confirm that the specific conditions in \[2\] are satisfied by the method proposed in \[1\]. We have already checked that the results in \[2\] hold for \[1\], but we do not know whether any paper has included such a result. To add probabilistic bounds to Theorem 2, one also needs to connect the results in \[2\] to the Koopman results, which is why we left this for future work.
>
> 2. Yes, all the data is used for the approximation of $K$ in all experiments, including the real-world data experiments. We agree that for very large $N$ and $M$, the computational intensity increases, which we also discuss as a limitation of our approach in the conclusion. In the experiments, we did not encounter scalability issues. For future work, a subsampling strategy or use of efficient linear solvers could mitigate computational challenges.
>
> 3.  We agree that the discrepancy between repeated application of the Koopman operator versus a repeated projection + re-lifting should not go unnoticed. In the last paragraph of Section 2 we discuss our current understanding of the two approaches, supported by recent literature. We do think that showing similar results in general for the repeated projection + re-lifting can be challenging, but with certain assumptions on the underlying manifold  $\\mathcal{F}\_M(\\mathcal{H})$, one might investigate the discrepancy further in the future.
>
> 4. This is an interesting question that also piqued our interest, and we could empirically confirm that the incorporation of the Koopman operator does indeed improve accuracy. For some insight into this, we refer to the last paragraph of Section 3.2 and, in particular, the results shown in Table 2.
>
> **Answers to minor comment:**
>
> Thank you for this correction. We have removed $ \\mathcal{F}\_M (H’) $ in line 6 of Algorithm 2.
>
> **References:**
>
> \[1\] Bolager, Erik L and Burak, Iryna and Datar, Chinmay and Sun, Qing and Dietrich, Felix. “Sampling Weights of Deep Neural Networks”. Advances in Neural Information Processing Systems. 2023.
>
> \[2\] Rudi, Alessandro and Rosasco, Lorenzo. "Generalization Properties of Learning with Random Features”. Advances in Neural Information Processing Systems. 2017.

---

### Review · Reviewer_F3iR · 2026-04-28

**Summary Of Contributions:**

The paper introduces Koopman-informed recurrent neural networks (KIRNN). It constructs RNNs by randomly sampling hidden layer parameters using a data-dependent distribution based on Bolager et al. (2023), then approximates the Koopman operator via extended dynamic mode decomposition (EDMD) to obtain linear operators for state evolution. The approach is evaluated on synthetic ODEs , crowd dynamics, controlled systems using LQR, and real-world time series. The authors demonstrate quicker training times and achieve accuracy comparable to other methods.

**Additional Comments:**

The main technical contribution is novel but undermined by avoidable presentation and consistency problems. The speed advantage appears real and valuable.

**Audience:**

Yes

**Audience Explanation:**

Yes. The combination of data-dependent random feature sampling for RNNs with direct Koopman operator approximation offers a distinct perspective on efficient training and interpretability for dynamical systems.

**Claims And Evidence:**

No

**Claims Explanation:**

1. Algorithm-equation inconsistency: Algorithm 2 defines C = H F_M(H') F_M(H)+, which does not match the earlier derivation C = H F_M(H)+. This is a clear technical error.

2. Unclear core derivation: Koopman operator theory is introduced in Section 2.2 after the sampled RNN is already defined. This creates a break in the method description. Equation (7) is presented abruptly without a clean step-by-step mapping from Definition 1 and the earlier C_h, C_x formulation.

 3. Mixed empirical support for accuracy: Table 1 shows KIRNN is faster in all cases, but accuracy is only sometimes better than ESN or shPLRNN (e.g., worse than ESN on Rössler attractor under EKL). The narrative of consistent superiority is not fully supported.

 4. Weak control evaluation: The controlled Van der Pol experiment demonstrates LQR feasibility but provides no competitive baseline (explicitly noted by the authors). The claim of a meaningful advantage for nonlinear control, therefore, lacks convincing comparative evidence.


5. Theory-practice mismatch: Theorem 2 and Assumption 1 are for the uncontrolled case and exclude ReLU, yet experiments use ReLU for crowd dynamics, and the theory section does not clearly delineate these limitations in the main text.

6. Missing comparison to neural operator models:  The work targets dynamical systems but there is no comparison to neural operators (standard for PDE/continuum modeling). Stronger baselines that should have been included are: Fourier Neural Operator (FNO), DeepONet, Neural Operators with Koopman-inspired lifting. These would provide a more rigorous benchmark, especially for long-horizon forecasting and PDE-style problems

**Requested Changes:**

- Correct the inconsistency in Algorithm 2 for matrix C and verify all pseudocode against the mathematical derivations.
- Restructure Section 2 to introduce the Koopman operator before defining the final model, followed by an explicit derivation of Equation (7).
- Revise accuracy-related claims to reflect the mixed results shown in Table 1.
- Add at least one strong baseline for the control experiments and report quantitative comparison metrics.
- Add stronger baselines: auto-regressive neural operator based models

---

> ### Author Response · Authors · 2026-05-08
> **Response to Reviewer F3iR**
>
> We are thankful for the provided feedback. In this response we aim to answer the reviewer’s questions and address their concerns. We have updated the manuscript and marked the changes with blue. If some questions remain open please let us know during the discussion phase.
>
> **Answers to major comments:**
>
> 1. Thank you for pointing this out. We have corrected Algorithm 2 in the updated manuscript.
>
> 2. We agree that the presentation in Section 2 can be improved. We have restructured Section 2.2 such that we first introduce the EDMD algorithm and then show how this can be incorporated into the sampled RNN scheme.
>
> 3. The main interpretation of Table 1 should rather be that our method performs comparably and occasionally better than other models in terms of accuracy; this was our intention when writing Section 3. In our view, KIRNN can provide computational benefits and enable fast hyperparameter tuning, while being comparable in accuracy. We can see that when discussing both computational benefits and accuracy in the same sentence, this can be confused with claiming superiority in both domains. To address your concerns we have revised the text and have made adjustments to our claims to better reflect the quantitative results.
>
> 4. In the control experiment we show that our method is compatible with existing tools from linear control theory; for RNNs in general this is not the case, thus a comparison with an RNN approach is not possible. We consider the compatibility with linear control theory to be an advantage from a computational aspect because linear controllers are typically less expensive and easier to work with than nonlinear ones, this is well established in the literature. Furthermore we do not claim that our control approach is superior to nonlinear methods, such as the approaches proposed in \[1\] and \[2\], however these methods do not fall under the RNN category thus we do not consider a comparison with a starkly different modeling approach.
>
> 5. It is true that in the main paper we do not explicitly state that Assumption 1 is not satisfied for the ReLU activation function; this is discussed in the appendix. We have now revised Section 2.4 to mention the incompatibility with ReLU. For the crowd experiment, we tried both tanh and ReLU as activation functions, and with tanh the results were satisfactory, but we decided to use ReLU in the final version because the results were slightly better than with tanh. It is not unusual for empirical evidence to sometimes deviate from the assumptions of the theoretical work. In our view, the fact that Theorem 2 would provide support for a model using a tanh activation should not restrict users to consider tanh as the only option, but they should have the flexibility to choose any activation function for their problem. In all but one experiment, we found that tanh is an excellent choice.
>
> 6. Neural operators and auto-regressive neural operators extend traditional neural networks to the domain of functions (i.e., mapping from a function space to a function space), hence they are suitable for PDEs.
> In all the systems we consider, the domain is a Euclidean space, therefore using neural operators is not suitable in these cases. The reason our gradient-based baseline is shPLRNN is that it has been shown to outperform other relevant architectures in this finite-dimensional setting, particularly for chaotic systems where shPLRNN is also strongly theoretically motivated. We further compare KIRNN with reservoir models because they are suitable representatives of random-feature models and offer good computational benefits. In essence, the models chosen for comparisons are well-motivated, reasonably chosen, and thus provide strong baselines.
> For future work, our approach can be extended to the infinite-dimensional setting, including PDE problems, for example by using our sampling approach to train autoregressive neural operators. In that case, we certainly agree that neural operators will provide a suitable baseline for comparison.
>
> **References:**
>
> \[1\] Halás, M., & Dodek, M. “Realization of proper nonlinear systems and its application to the disturbance observer for Van der Pol oscillator”. IEEE Access. 2025
>
> \[2\] Alghassab, M., Mahmoud, A., & Zohdy, M. A. “Nonlinear control of Chaotic Forced Duffing and Van der Pol oscillators”. International Journal of Modern Nonlinear Theory and Application. 2017

---

### Decision · Action_Editor_tggc · 2026-06-07

**Recommendation:** Accept with minor revision

**Additional Comments:**

Two of three reviewers recommended acceptance (leaning accept), while Reviewer F3iR recommended rejection primarily on grounds of missing baselines and presentation issues. The authors submitted a rebuttal and revised manuscript addressing the main concerns.

The most concrete technical issue — the inconsistency in Algorithm 2's definition of C — was corrected. The authors also restructured Section 2.2 to present EDMD before the Koopman factorization, improving the narrative flow. The accuracy claims were softened to better reflect the mixed results in Table 1, which is the right call. Reviewer F3iR's insistence on neural operator baselines (FNO, DeepONet) is not compelling here since all experiments involve finite-dimensional ODE/time-series problems, not function-space mappings; the authors' response on this point is reasonable.

The remaining minor revision needed is primarily around clarifying the role of hidden states h in Algorithm 2 — Reviewer NYv7 raised this and the authors added a paragraph, which the reviewer found helpful but not fully satisfying. The manuscript should make explicit, early in Section 2, that h corresponds to the full state (or time-delay embeddings thereof) and is treated as observed during training, distinguishing this from the typical RNN setting where h is latent. This is a presentation fix, not a fundamental flaw.

The control experiment remains somewhat thin — no quantitative comparison to any baseline — but the authors' framing (demonstrating LQR compatibility as a structural advantage of the Koopman factorization) is defensible as a proof-of-concept. The minor revision should at minimum add a clearer disclaimer about the scope of this experiment's claims.

**Audience:**

Yes

**Audience Explanation:**

Researchers working on dynamical systems modeling, Koopman/EDMD methods, and efficient RNN training will find this work relevant. The idea of constructing RNN weights without gradient-based optimization — using random features plus a least-squares solve — is a practically useful alternative to BPTT that the community can learn from. The explicit connection drawn between EDMD and RNN architectures is also of interest to both communities.

**Claims And Evidence:**

Yes

**Claims Explanation:**

The core claims — that KIRNN avoids BPTT-related gradient issues by combining random feature sampling with EDMD, and achieves competitive accuracy with faster training — are reasonably well supported. Two of three reviewers found the evidence convincing. The main empirical concern raised by Reviewer F3iR was that Table 1 shows mixed accuracy results (e.g., KIRNN underperforms ESN on Rössler under EKL), and the authors appropriately revised their claims to reflect 'comparable' rather than consistently superior accuracy. The Algorithm 2 inconsistency (the C matrix definition) was a concrete technical error that the authors corrected in revision. The control experiment lacks a competitive baseline, which is a genuine gap, but the authors reasonably argue that linear control compatibility is the point being demonstrated rather than superiority over nonlinear methods. Reviewer F3iR's push for neural operator baselines is not well-motivated given that the benchmarks are finite-dimensional ODE systems, not PDE function-space problems.